# Regulation of developmental gatekeeping and cell fate transition by the calpain protease DEK1 in *Physcomitrium patens*
Viktor Demko[1,7,8], Tatiana Belova[1,9], Maxim Messerer [2], Torgeir R. Hvidsten [3], Pierre-François Perroud [4], Ako Eugene Ako[5,10], Wenche Johansen[5], Klaus F. X. Mayer[2,6], Odd-Arne Olsen [1] & Daniel Lang [2,11] ✉

Calpains are cysteine proteases that control cell fate transitions whose loss of function causes severe, pleiotropic phenotypes in eukaryotes. Although mainly considered as modulatory proteases, human calpain targets are directed to the N-end rule degradation pathway. Several such targets are transcription factors, hinting at a gene-regulatory role. Here, we analyze the gene-regulatory networks of the moss *Physcomitrium patens* and characterize the regulons that are misregulated in mutants of the calpain DEFECTIVE KERNEL1 (DEK1). Predicted cleavage patterns of the regulatory hierarchies in five DEK1-controlled subnetworks are consistent with a pleiotropic and regulatory role during cell fate transitions targeting multiple functions. Network structure suggests DEK1-gated sequential transitions between cell fates in 2D-to-3D development. Our method combines comprehensive phenotyping, transcriptomics and data science to dissect phenotypic traits, and our model explains the protease function as a switch gatekeeping cell fate transitions potentially also beyond plant development.

Multicellular organisms establish distinct cellular identities leading to individual tissues, cell types and functions when cells acquire specific cellular fates in response to environmental signals and developmental patterning cues[1]. Transitions between cellular fates are accompanied by reprogramming of gene expression and modulation or turnover of the cell protein complement. In plants, cell fate integrates spatial localization and intercellular communication in a highly coordinated manner. In particular, asymmetric, formative cell divisions mostly involve a reorientation of the division plane[2].

Aleurone cells of the grain endosperm and the protodermal and derived epidermal L1 layers of flowering plants are prime examples for cell fate specification. These cell types form the outer layers of the respective flowering plant tissues via asymmetric cell division as a function of their

position[3,4]. Notably, loss of DEFECTIVE KERNEL 1 (DEK1) function in monocots[5] and dicots[6] abolishes these cell fate specifications, with strong *dek1* alleles causing embryo lethality. *dek1* mutants in various organisms display pleiotropic phenotypes with severely impaired development, including that of the shoot apical meristem[6,7]. Consistent with a role for DEK1 as a key factor in the evolution of land plant meristems[8], studies in the bryophyte model system *Physcomitrium patens*[9] demonstrated a vital role for DEK1 as a developmental regulator controlling cell fate decisions in the moss simplex meristems during their transition from 2D to 3D growth[10,11].

Null *dek1* mutants have disorganized division planes resulting in defective division patterns and cell shapes[6,10] due to defects in microtubule-mediated orientation, cell wall deposition and remodeling, and cell adhesion[12]. Changes in gene expression levels and patterns also hint toward

[1]Department of Plant Sciences, Norwegian University of Life Sciences, P.O. Box 5003, NO-1432 Ås, Norway. [2]Plant Genome and Systems Biology, Helmholtz Center Munich—Research Center for Environmental Health, 85764 Neuherberg, Germany. [3]Faculty of Chemistry, Biotechnology and Food Science, Norwegian University of Life Sciences, Ås, Norway. [4]Institut Jean-Pierre Bourgin, INRAE, AgroParisTech, Université Paris-Saclay, 78000 Versailles, France. [5]Department of Biotechnology, Inland Norway University of Applied Sciences, Holsetgata 31, 2318 Hamar, Norway. [6]School of Life Sciences, Technical University Munich, 85354 Freising, Germany. [7]Present address: Department of Plant Physiology, Faculty of Natural Sciences, Comenius University in Bratislava, Ilkovicova 6, 84104 Bratislava, Slovakia. [8]Present address: Plant Science and Biodiversity Center, Slovak Academy of Sciences, Dubravska cesta 9, 84104 Bratislava, Slovakia. [9]Present address: Centre for Molecular Medicine Norway, University of Oslo, Oslo, Norway. [10]Present address: School of Animal, Rural and Environmental Sciences, Nottingham Trent University, Brackenhurst Campus, Southwell, Nottinghamshire NG25 0QF, UK. [11]Present address: Bundeswehr Institute of Microbiology, Microbial Genomics and Bioforensics, 80937 Munich, Germany. ✉e-mail: daniel.lang@mailbox.org

a regulatory role for DEK1, as the expression of several cell-type-specific transcription factors including ML1, PDF2, HDG11, HDG12, and HDG2, embryo- and post-embryo developmental regulators such as CLV3, STM, WOX2, WUS, PIN4 and potential downstream target genes involved in cell wall biosynthesis and remodeling including XTH19, XTH31, PME35, GAUT1, CGR2, EXP11 are all misregulated[11–13].

DEK1 is a 240-kDa multi-pass transmembrane (TM) protein with a cytosolic calpain cysteine protease (CysPc and C2L domains) as effector (Fig. 1a)[11,14]. DEK1-type calpains are deeply conserved and likely evolved in eubacteria[15]. Calpains constitute a third proteolytic system besides the lysosomal and proteasomal systems[16]. $Ca^{2+}$-dependent calpains are considered modulatory proteases that cleave proteins at a few specific sites, generating fragments or neo-proteins with novel functions (e.g., activating preproteins) or modulating protein function, associations and localization. Like plant DEK1 calpains, classical calpains are pivotal to animal development and cell fate transitions, suggesting that the ancestral functions of the calpain superfamily are cell division and cell cycle regulation[17]. Human calpains are aggravating factors in many pathophysiological conditions and illnesses, including cancers and hereditary diseases like muscular dystrophy[17,18].

Calpains have fuzzy target specificity that is less reliant on the primary sequence of the substrate and may depend on higher-order factors like 2D or 3D protein structure or cofactors[17], complicating the systematic identification of calpain targets. Indeed, a few hundred calpain substrates have been reported, mostly in mammals. Notably, no direct DEK1 substrates are known in plants, which is at odds with the severe impact of loss of calpain function.

Mammalian calpain targets are short-lived substrates for the N-end rule degradation or N-degron pathway (NERD)[19–21]. These proteins bear N-terminal residues (N-degrons) that attract and activate the NERD pathway, leading to their ultimate degradation by the ubiquitin-proteasome system[19]. Here, calpain cleavage causes destabilization or inhibition of biological functions, providing a plausible explanation to align calpains' limited target specificity with their broad biological effects. However, as calpains also target transcription factor (TFs) and transcriptional regulators, a more direct path to impact both the physical and regulatory layers of cell fate transitions emerges. In this case calpains act as post-translational regulators of gene functions through either (a) directly modulating protein function; (b) indirectly affecting the stability of target proteins by marking them for the NERD pathway; or (c) indirectly controlling the stability of transcriptional regulators (Fig. 1e). While (a) provides a positive control over a gene's function, (b) an inhibitory, negative control, the outcome of (c) depends on whether the targeted TF or regulator represents a transcriptional activator or repressor and thus offers bidirectional control of gene functions.

Due to its subcellular localization in the plasma membrane and its role in the establishment of cell division plane orientation, so far studies of the plant calpain have been designed with a scenario in mind where DEK1 only acts as a modulatory protease targeting a limited number of specific targets in or around the cell division plane apparatus. Combining the pleiotropy and developmental essentiality in DEK1 mutants, the fuzzy target specificity and the broad range of functions with the inhibitory role in targeting proteins to the NERD pathway, we here postulate a second scenario (dual role scenario), where in one role the calpain directly modulates specific protein functions and in a second role, it indirectly controls the half-life of potentially many proteins via the NERD pathway. As this latter role may also affect the protein stability of transcriptional regulators, it could affect the expression of a large number of target genes and thereby help to understand calpains' widespread developmental and gene-regulatory impact.

The model moss *P. patens* provides an ideal system in which to dissect these scenarios, due to its evolutionary position, high-quality reference genome and annotation[22], well-established molecular toolbox and comprehensive transcriptomics resources (e.g. refs. 23,24). In particular, the simple morphological structure during early moss development, comprising mostly single-cell-layered tissues, is not impeded in null *dek1*

mutants[10,11,14], allowing a comprehensive dissection of 2D to 3D transition in the moss simplex shoot meristem.

We set out to elucidate the position of DEK1 during plant development and transcriptional regulation by combining phenotypic and transcriptome profiling of wild-type (WT) and mutant moss lines with a comprehensive, genome-wide, integrative, multi-scale data-mining approach to analyze the misregulation of global gene-regulatory networks (GRNs) in the mutants. The predicted network supported a function for calpain as a post-translational regulator of gene expression. Importantly, the proposed model explains the protease's role as a developmental switch gatekeeping cell fate transitions.

## Results

### Loss of *DEK1* dramatically affects moss development

Previous work demonstrated that *dek1* mutations dramatically affect *P. patens* development[7]. Here, we characterized the phenotypes and transcriptome profiles of five moss lines: WT; a *Δdek1* deletion strain[10]; a strain accumulating the DEK1 linker and calpain domains whose encoding sequence was driven by the maize (*Zea mays*) *Ubiquitin* promoter (*oex1*, Fig. 1a); and two lines carrying partial *DEK1* deletions, *dek1Δloop*[11] and *dek1Δlg3*[14] (Fig. 1a). We collected samples at five time points comprising the early and intermediate stages of *P. patens* development, including the transition from 2D tip growth in filamentous protonema to 3D apical growth in leafy shoots (gametophores; Fig. 1b). Although early protonema development was largely unaltered in the *Δdek1* and *dek1Δloop* mutants (Fig. 1b), later stages, including the gametophore formation, were substantially disturbed in all mutant strains.

The *Δdek1* and *oex1* lines exhibited opposite phenotypes compared to WT: reduced (*Δdek1*) or enhanced (*oex1*) secondary filament extension, higher or lower percentage of filaments forming buds (*Δdek1*, *oex1*) and a four-fold higher gametophore bud initiation rate per filament in *Δdek1* (Fig. 1b, c). The partial *dek1Δloop* deletion line displayed milder phenotypic changes than *Δdek1*, except for bud development, as the moss continued to proliferate and form naked stems without initiating phyllids. dek1Δlg3 showed unique phenotypes, including severely affected protonema differentiation and branching resulting in reduced plant size, and aberrant gametophore formation. Juvenile dek1Δlg3 plants had stunted phyllids.

How can altering one membrane-bound protease have such variable, complex and drastic effects on plant development? The clear, distinct phenotypes of *dek1* lines provided an opportunity to explore this question through transcriptome deep sequencing (RNA-seq).

### Many genes and functions are misregulated in *dek1* mutants

We performed differential gene expression (DGE) analysis based on triplicate RNA-seq libraries generated for the above developmental time course in all five lines, testing all protein-coding and non-coding genes for DGE using kallisto/sleuth[25], which identified sets of upregulated and downregulated genes in the mutants at a false-discovery rate (FDR) of 10% and at 1% (Fig. 1d and Supplementary Data S1). Detailed analysis of gene sets inferred using both filtering criteria (Supplementary Data S1), as well as comparison with those obtained by an alternative DGE method and existing experimental data[11] suggested relaxed FDR cut-off ($q$ value < 0.1) as an optimal tradeoff between false positives and false negatives for the subsequent multi-step data analysis procedure to elucidate the global impact of dek1 mutation.

Consistent with the dramatic phenotypic consequences of *Δdek1* and *oex1*, we detected the largest number of misregulated genes between *Δdek1* (35% of all genes) and *oex1* (44% relative to WT and 49% relative to *Δdek1*). Only ~7% of all genes were misregulated in the *dek1Δloop* and *dek1Δlg3* lines. Notably, we observed balanced misregulation, with comparable numbers of upregulated and downregulated genes.

Overall, the extent of misregulated genes supported a dual-role scenario for DEK1. The balanced directionality of misregulated genes also suggested that DEK1 cleaves activators and repressors equally. To delineate

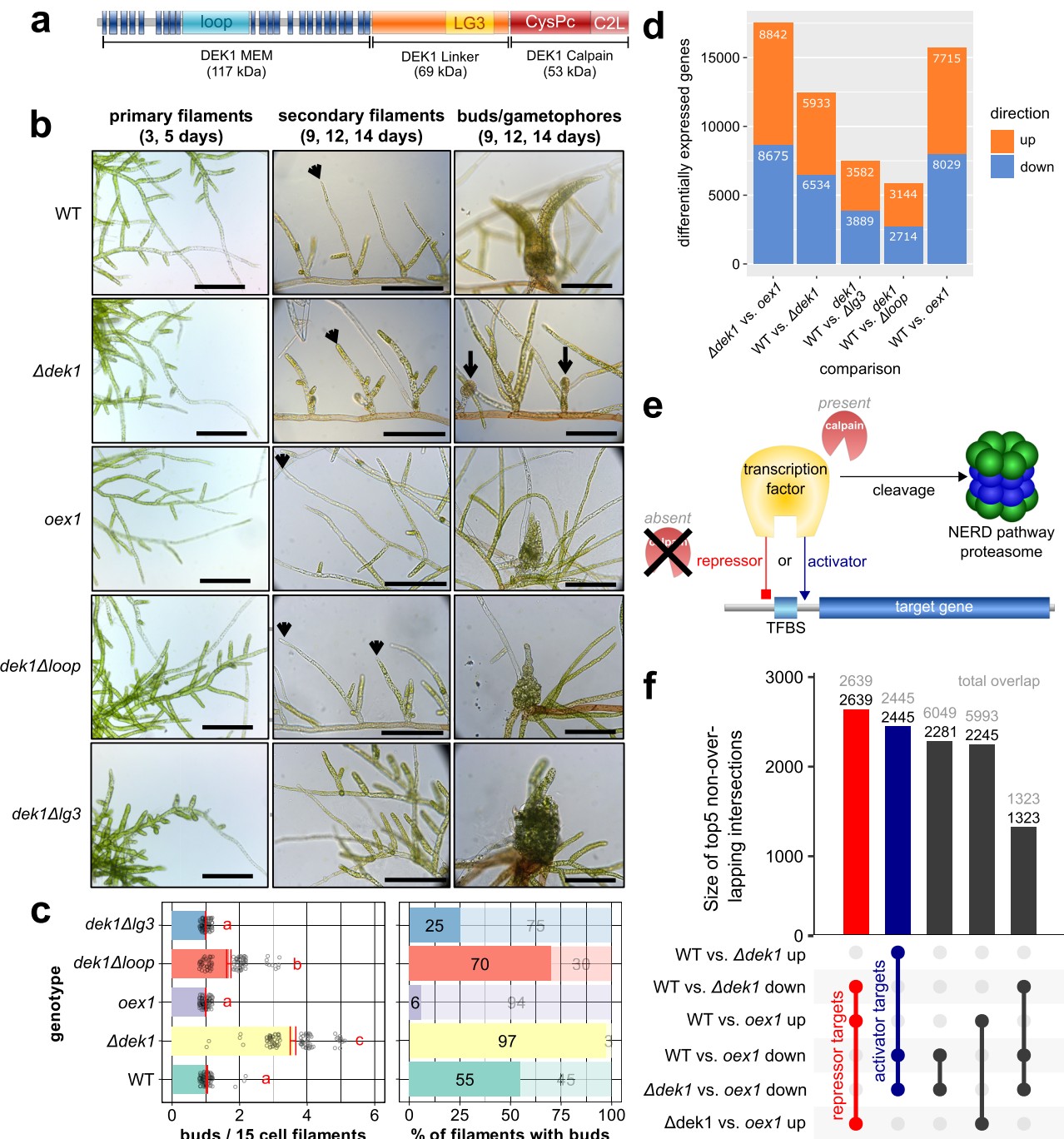

**Fig. 1 | Phenotypes and transcriptome profiling of *dek1* mutant lines in *P. patens*.**
**a** DEK1 protein domain structure. **b** Time series analysis of *P. patens* juvenile gametophyte development in WT, a DEK1 calpain domain overexpressor (*oex1*), a complete deletion of the *DEK1* gene (*Δdek1*)[10] and two partial deletion lines lacking the loop (*dek1Δloop*)[11]; or the LG3 domain (*dek1Δlg3*[14]). Microscopy images (scale bars: 200 μm) show primary filaments in early stages of protonemata development (3, 5 days), secondary filaments (9–14 days; arrowheads point to apical cells of individual secondary filaments), buds and gametophores (9–14 days; arrows point to arrested buds in *Δdek1*). **c** Quantitative analysis of gametophore apical stem cell (bud) formation in *dek1* mutants (*y*-axes both panels: color-coded genotypes). The frequency of meristem initiation is expressed as mean number of buds per 15-cell-long filament (left panel, *n* = 100) and percentage of filaments forming buds (right panel, *n* = 100). Statistical significance at 95% confidence is indicated in left panel for mean number of buds (red annotation: **a**–**c**). Analysis of variance (ANOVA) and least significant difference (LSD) test were performed in multiple sample comparisons. Individual black open circles in left panel indicate individual data points. Red error-bars in left panel indicate standard errors. Lighter colored bars (alpha

transparency) in right panel indicate percentages of filaments without bud.
**d** Pairwise differential time series gene expression analysis of *dek1* mutants at 3, 5, 9, 12 and 14 days. Stacked bar chart of significantly differentially expressed genes (DEGs) with a false discovery rate (FDR) < 0.1. Orange, upregulated genes; light blue, downregulated genes. **e** Working model for the gene-regulatory role of DEK1/ calpains. When active calpain is present, a TF is cleaved and targeted to the N-end rule degradatory (NERD) pathway, resulting in loss of gene regulation. In the absence of active calpain, the TF regulates target gene expression either as an activator (blue) or repressor (red). **f** Size of the top 5 intersections between the DEG sets in (**d**). Numbers above bars depict the proportion unique to the given set (black) as well as the total (gray) size of each intersection. The largest set (red) comprises 2639 genes, which are downregulated in *Δdek1* and upregulated in *oex1*, making these genes targets of DEK1-controlled repressors. The second-largest set (blue) comprises 2445 genes that are upregulated in *Δdek1* and downregulated in *oex1*, likely controlled by DEK1-targeted activators. These sets represent the most conservative lists, as the third intersection likely contains additional activator targets with weak FDR support in the comparison of *Δdek1* and the WT.

the functional consequences of *dek1* mutations, we used Gene Ontology (GO) and Plant Ontology (PO) annotations to assess the global functional impact of the misregulated genes in the mutants (Supplementary Fig. S1 and Supplementary Data S2). Consistent with a dual role for DEK1 and the observed pleiotropic phenotypes, we determined that 85% of molecular functions, 88% of biological processes, 90% of cellular components, 92% of anatomical entities and 94% of all developmental stages in GO and PO are misregulated in *Δdek1*.

### Potential indirect DEK1 targets show consistent mutant mis-regulation patterns

If DEK1 is a post-translational regulator (Fig. 1e) of ubiquitous gene functions, the RNA-seq datasets should allow us to identify the targets of repressors cleaved by DEK1, as their expression should be upregulated in the *oex1* line and downregulated in *Δdek1* (referred to hereafter as repressor targets); genes downstream of DEK1-controlled activators should exhibit the opposite pattern (activator targets). We thus performed multiple comparisons between lines to identify differentially expressed genes (DEGs) with the predicted misregulation patterns (Fig. 1f). Indeed, the largest set, of 2639 genes (red bar; Fig. 1f), was downregulated in *Δdek1* and upregulated in *oex1*, marking these genes as targets of DEK1-controlled repressors (repressor targets) (Fig. 1e). The second-largest set (blue bar; Fig. 1f) comprised 2445 genes upregulated in *Δdek1* and downregulated in *oex1*, suggesting that these genes are targets of DEK1-controlled activators (activator targets) (Fig. 1e). The third and fourth most frequent patterns were subsets of these two gene sets. Both gene sets also likely included genes under the indirect control of DEK1. In subsequent analyses, we focused on the two major sets of consistently misregulated genes as potential indirect targets of repressors and activators controlled by DEK1 (Fig. 1f).

During WT gametophyte development, 71% of the putative repressor target genes and 75% of putative activator target genes exhibited substantial changes in expression levels. The two sets differed in their expression patterns over the time course, as 765 repressor targets were more highly expressed during the early phase (days 3–5, Fig. 1b), while 794 activator targets were upregulated during later development (days 9–14). This suggested that some potential DEK1 targets are involved in the developmental transitions occurring during this period.

### Gene regulatory subnetworks are enriched for putative DEK1 targets

We next looked for any consistent misregulation patterns in the moss GRNs. To this end, we compiled 374 public and novel RNA-seq libraries and 1736 novel annotated regulators (see Supplementary Data S2 for full information) using the random forest predictor of GENIE3[26] and calculated Pearson's correlation coefficients between regulator and target gene expression levels. We then detected the top 10 regulatory interactions for 35,706 genes, which resulted in 11 robust subnetworks (Supplementary Fig. S2a). We used these predicted regulatory interactions and subnetworks as tools to assess the putative gene-regulatory role of DEK1.

Using the candidate DEG sets, we performed network enrichment analysis for the identified subnetwork graphs[27], finding significant enrichment of regulatory relationships of DEK1-controlled repressor and activator targets in subnetworks II, V, VIII, IX and X (FDR ≪ 0.01; Fig. 2a, rows 1, 3 and Supplementary Fig. S2b). Subnetwork V appeared to be enriched for repressor targets that are active during the early phase of moss development consisting mostly of chloronema filaments (3–5 days; Fig. 2a, row 2). Subnetworks II and X were enriched for activator targets expressed during the 2D-to-3D growth transition (9–14 days; Fig. 2a, row 5). Subnetwork IX encodes housekeeping gene functions including primary gene regulation, transcription, translation, constitutive epigenetic regulation as well as light-independent mitochondrial and cytosolic metabolic pathways (Fig. 2b, e). Subnetwork VIII harbors the light-dependent and -responsive pathways, in particular photosynthesis, plastid-morphogenesis/regulation and generally plastid-localized pathways (Fig. 2b, e).

By tracing the regulatory links of misregulated genes in the GRN, we identified potential upstream TF genes with unaltered expression levels in the mutants but whose protein products may be direct cleavage targets of DEK1. Consequently, we tested subnetworks for enrichment of such upstream TFs predicted to directly control any of the misregulated genes. This analysis highlighted subnetworks II, VIII and X, but also suggested potential direct cleavage targets in three other subnetworks (I, III and XI; Fig. 2a, row 4). As the misregulated targets of these TFs predominantly also fell into the five DEK1-controlled subnetworks, the latter group of regulators might serve as DEK1-controlled interfaces to other regulatory circuits.

### Network structure suggests DEK1-gated sequential transition between cell fates

The majority of regulatory interactions are found within subnetworks (Supplementary Fig. S2a). However, as evident from the enrichment of inter-subnetwork connections (Supplementary Fig. S2a), the putative indirect DEK1 targets are predicted to be also controlled by regulators from other subnetworks. In order to determine whether the respective TFs act as positive (activators) or negative (repressors) regulators of potential DEK1-controlled gene functions, we studied the directionality of the inter-subnetwork connections based on the sign of the expression profiles' correlation coefficients (black [+] vs. red [−] colored edges in Fig. 2b and Supplementary Fig. S2c and Supplementary Data S6) and compared the relative proportions of intra- and inter-connections among the enriched subnetworks split according to positive and negative interactions (Figs. 2b, f and S4).

We found more than expected negative links between V ⊣ II and X ⊣ V (i.e., TFs from V repressing targets in II and X TFs as repressors of V targets) and more positive, activating regulatory interactions between II → X and X → II (i.e., TFs from II activate genes in X and vice versa; Fig. 2f). In our interpretation, this chained pattern potentially reflects the developmental transition between different cell fate identities including primary filament cells differentiating to side branches and gametophore buds. Furthermore, we also observe a biased distribution of activator and repressor targets among the subnetworks (Fig. 2a). While subnetwork IX contains both patterns, V harbors more repressor targets and II, VIII and X comprise more activator targets. It therefore seems that the enriched subnetworks respond in a specific fashion to the mutation of the plant calpain DEK1.

Taken together, the preferential developmental timing and the biased directionality chain suggest the presence of an inherent directionality of DEK1 action on the regulatory circuitry of these subnetworks. These findings may hint toward a mechanism in which DEK1 affects distinct cellular identities as displayed in the DEK1 mutants.

### DEK1 is part of the APB-controlled subnetwork II guarding the 2D-to-3D transition

The directed edges of the GRN graph can be used to reconstruct a regulatory hierarchy by ranking regulators according to their network centrality. Applying this local reaching centrality criterion, we identified the TF AINTEGUMENTA, PLETHORA and BABY BOOM 4 (APB4) as the master regulator at the top of the regulatory hierarchy in subnetwork II (rank 1; reaching >99% of the subnetwork). The eponymous members AINTEGUMENTA, PLETHORA and BABY BOOM of this subfamily are involved in various developmental processes including the formation of the stem cell niche in the Arabidopsis shoot apical meristem[28,29].

Consistent with the documented role as master regulators of moss gametophore apical stem cell formation, the timed tissue and cell-type specific expression patterns and the additive, but distinct phenotypic severity of single, double, triple and quadruple knockout mutants of the four moss APBs[30], the outparalogous copies APB2 (rank 35) and APB3 (rank 41) are localized downstream of APB4 in the regulatory hierarchy of subnetwork II. The inparalogous APB1 is localized in subnetwork VII, but is also an indirect target of APB4 (Supplementary Fig. S5a). DEK1 is predicted to be localized downstream of APB2 and APB1 in subnetwork II (Supplementary Fig. S5b).

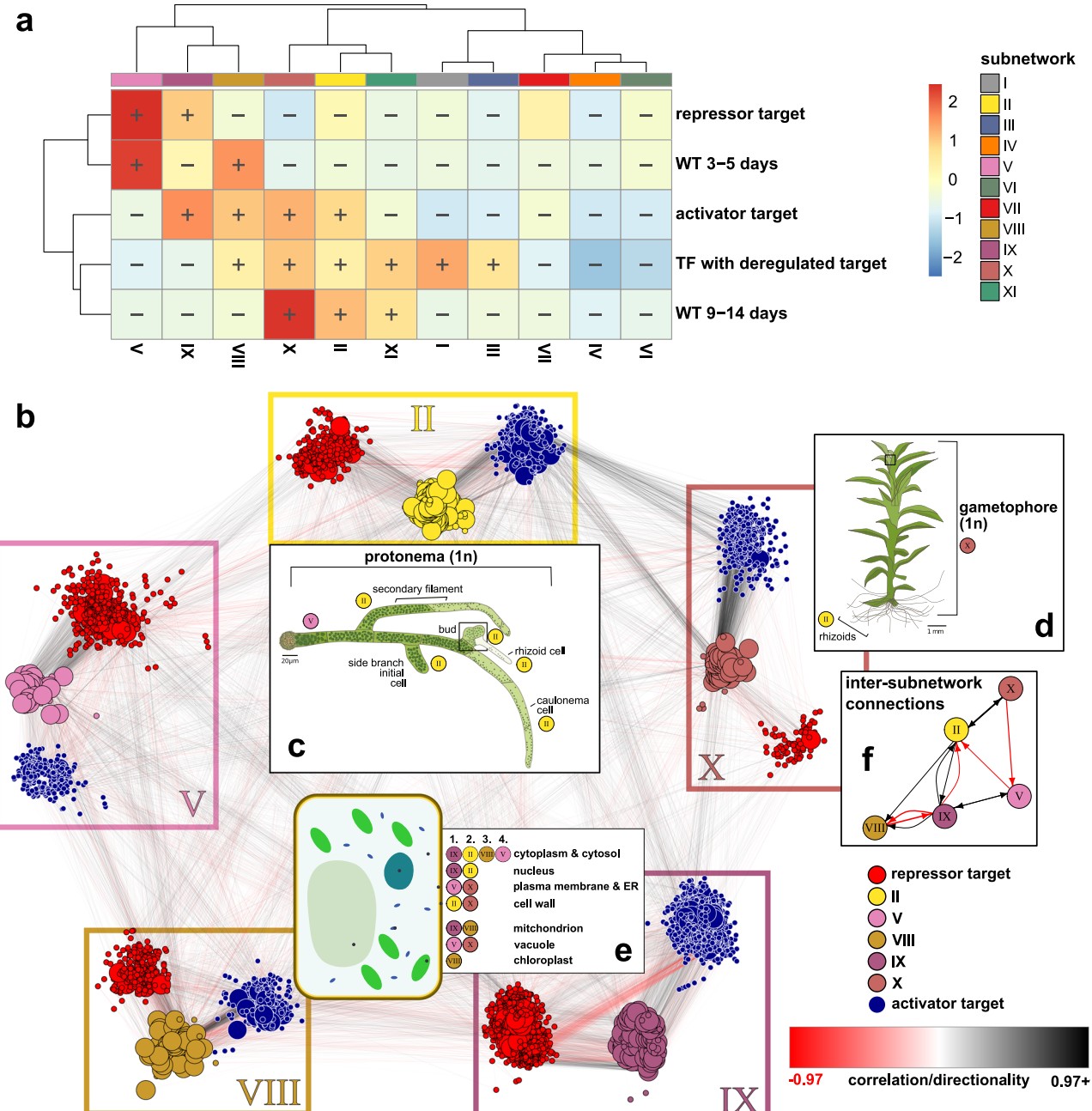

**Fig. 2 | Tracing DEK1-misregulated genes and their upstream regulators in the predicted *P. patens* gene-regulatory network (GRN) highlights subnetworks that encode the developmental transitions governed by DEK1.** Prediction of regulatory interactions with subsequent clustering results in 11 subnetworks (Supplementary Fig. S2a). **a** Network enrichment analysis highlighting specific overrepresented subnetworks among DEK1-misregulated gene sets (activator and repressor targets) and their upstream regulators (TFs with misregulated target) as well as the distinct phases of WT development (WT 3–5 days and 9–14 days). See Fig. S2d for overlap analysis of these gene sets. Heatmap represents the ratio between observed and expected sizes of specific candidate gene sets among the identified subnetworks. Significant (FDR < 0.01) enrichment (+) or depletion (−) is shown. Ratios were clustered for both rows and columns using the ward.D2 method. **b** Network graph of the five DEK1-misregulated subnetworks: for each enriched subnetwork (II, V, VIII, IX and X), all genes with an activator (blue nodes) or repressor (red nodes) misregulation pattern in the *dek1* mutants (Fig. 1f) are shown as nodes together with unchanged, direct upstream TFs (using subnetwork color codes) as a triangular subgraph in subnetwork-color-framed boxes. Node sizes are scaled by local-reaching centrality, i.e., the fraction of the total subnetwork that can be reached via regulator → target connections. Edges, representing the predicted regulatory

interaction between a TF and its target, are colored according to putative directionality, with negative, repressive interactions in red and positive, activating regulatory interactions in black. Insets (**c**) and (**d**) significantly enriched developmental stages and tissue or cell types. **c** Schematic of the predicted roles of subnetworks V and II in the different cell fates comprising the haploid protonema stage. **d** Schematic of the haploid, leafy, juvenile gametophore that, except for the filamentous rhizoids encoded by subnetwork II, is predominantly implemented by subnetwork X. **e** Schematic of a plant cell depicting the significantly enriched intracellular localizations of DEK1-controlled subnetworks. In the accompanying text box, subnetworks are ranked (1–4) according to the percentages of genes with terms affiliated with the respective compartment. **f** Small network plot showing major significantly enriched inter-subnetwork connections (Pearson residuals > 4; Supplementary Fig. S4). Drawings of the *Physcomitrium* protonema (**c**) and gametophore (**d**) stages adapted from ref. 103. Drawing of plant cell (**e**) adapted from Wikimedia Commons User *Domdomegg*. Subnetwork assignments to developmental stages and cell types (**c**, **d**) are based on network enrichment of stage-specific DGE sets inferred from *Physcomitrium* gene atlas data[24]. Ranked subnetwork assignment of subcellular localizations (**e**) is based on ontology enrichment analysis of GO cellular component terms (Supplementary Fig. S3 and Supplementary Data S3).

The immediate upstream regulatory context suggests that DEK1 (Supplementary Fig. S5b) is positively regulated by subnetwork V TFs, activated early in development and subsequently negatively controlled by an auxin/ent-kaurene responsive cascade[31,32] that is encoded downstream of the aforementioned APBs by subnetwork II regulators as well as a hierarchical cascade of SQUAMOSA promoter binding protein-like (SBP) TFs in subnetwork X which already has been shown to be involved in bud formation[33]. In light of the biased directionality chain identified in the DEK1-controlled inter-subnetwork connections (Fig. 2f and Supplementary Fig. S2c), this might represent a negative feedback loop buffering the 2D-to-3D transition at the transcriptional level.

### Misregulation of GRNs is consistent with a role for DEK1 as a post-translational regulator

Mammalian calpains direct proteins toward the NERD pathway[19]. Thus, potential direct DEK1 targets should bear N-terminal amino acid residues marking them for ubiquitylation and subsequent degradation by the proteasome (Fig. 3a). Importantly, the NERD pathway components were recently identified in *P. patens* and mutants in key components found to arrest the 2D-to-3D transition[34]. We predicted calpain cleavage sites using the program GPS-CCD[35] and classified the identified proteins based on the number of putative DEK1 cleavage sites and the prevalence of NERD signatures in their resulting novel N-termini (Supplementary Fig. S6a–f). Strikingly, the three DEK1-controlled subnetworks encoding the 2D-to-3D transition (V → II → X) were among the five subnetworks enriched for such NERD-like calpain cleavage patterns (Fig. 3a) and also displayed the highest levels of overall gene misregulation among the five DEK1-controlled subnetworks (X < II < V < IX < VIII; Supplementary Fig. S6g). Targets of misregulated TF genes are more likely to be misregulated than genes downstream of non-misregulated TF genes. While all five subnetworks showed significant and positive correlations of misregulation between target genes and their direct upstream TF genes ($\rho = 0.1560189$; Kendall's rank and Pearson's correlation tests; $p < 2 \times 10^{-16}$), the individual trends for the subnetworks mirrored those of the overall gene misregulation levels and confirmed the notion that subnetworks X, II and V are most affected by the loss of DEK1 function (Fig. 3b).

Targets of potential DEK1/NERD-controlled TFs were consistently more misregulated than other genes. As the three subnetworks were also enriched for NERD-like calpain cleavage sites, we investigated the dependency of these patterns of target gene misregulation on putative DEK1 cleavages in their upstream regulons. Indeed, the misregulation levels of the indirect DEK1 target genes in subnetworks II, V and X were positively correlated with the percentage of putative, direct DEK1 targets among their upstream TF cascades (Fig. 3c). The upstream regulons of misregulated genes were significantly enriched for TFs that are indirect and direct targets of DEK1 (87%; Pearson residuals $\gg 4$; $\chi^2$ test $p < \times 10^{-16}$, Fig. 3d). The regulatory cascades demonstrated consistent misregulation patterns. The direct upstream regulons (first order: TF → target) were mostly misregulated themselves, meaning that they either are indirect DEK1 targets because their upstream TF is controlled by DEK1 (Fig. 3d, lower left) or are both direct and indirect DEK1 targets. These TFs were directly cleaved by DEK1, and their expression was misregulated in the mutants because an upstream, higher-order TF was a direct DEK1 target (Fig. 3d, lower right). Consistently, second- (TF → TF→target) and third-order (TF → TF → TF→target) regulons of misregulated genes were enriched for predicted direct cleavage by a calpain (Supplementary Fig. S6h).

Filtering of the DEK1-controlled regulons to first-order interactions yielded 531 TFs that are predicted to be directly cleaved by a calpain (Fig. 3e and Supplementary Fig. S7a). These TFs were predicted to directly regulate the expression of 3679 significantly misregulated indirect DEK1 target genes (Supplementary Data S6). Eighty-five TFs were both potential direct and indirect DEK1 targets. These predicted first-order DEK1 targets formed a highly interconnected network, comprising 10,120 network edges (Supplementary Fig. S7b). Most of these genes (74%) were included in the five major DEK1-controlled subnetworks (Fig. 3e and Supplementary Fig. S7a). In addition, 73% of the inherent 4,082 inter-subnetwork connections targeted one of the five subnetworks (Supplementary Fig. S7c). More than half of these target genes in the three subnetworks were involved in the 2D-to-3D transition (V, II and X). We confirmed these results by ontology analysis, which suggested a clear functional delineation of biological processes implemented by the DEK1-controlled repressive and activating intra- and inter-subnetwork regulatory interactions (Fig. 3f, g and Supplementary Fig. S8a–e).

### Candidate targets suggest deep conservation of DEK1 control over plant development

Target genes from subnetworks X, II and V positioned downstream of DEK1-controlled activators were enriched in biological processes, cellular components and plant anatomical entities (color text; Fig. 3e–g and Supplementary Fig. S8a) that are directly linked to the observed *dek1* mutant phenotypes and tissue- and cell-type-specific expression profiles of *DEK1* in flowering plants and the moss[10–12,14,36,37]. The predicted DEK1-controlled genes regulated or comprised components determining cell polarity, axis, number, division, division plane and fate. For instance, these genes were involved in the biological processes regulation of asymmetric cell division (via *STRUBBELIG* orthologs[38]); callose deposition in cell walls and defined cellular components including the phragmoplast (via orthologs of *AUGMIN6*[39] and *TANGLED1*[40]) and cell plate (*STRUBBELIG* orthologs or *CLAVATA1b* [*CLV1b*][41]).

Besides these specific processes and compartments, the predominant pattern was consistent with the role of DEK1 as a regulator governing development. *dek1* mutants were impaired in general cell fate determination or transition and stem cell, meristem or primordium identity and initiation (e.g., shoot, flower, root and axillary bud meristems; development of endosperm, ovule and embryonic meristems; Fig. 3f). In addition, DEK1 target genes from subnetworks X, II and V included components were enriched in anatomical entities expected to be affected in *dek1* mutants (Fig. 3g), such as leaf lamina, epidermal cells[13] or meristem layer L1[7]. L1 provided two striking examples where flowering-plant orthologs of predicted indirect DEK1 targets are misregulated in *dek1* mutants of flowering plants: the homeobox domain leucine zipper IV TF gene *MERISTEM LAYER 1*[13] and *CLV3*[8].

Experimental evidence from *P. patens* pointed to an enrichment of the protonema side branch initial cell, and gametophore initial cell, two major, sequentially occurring cell types related to the 2D-to-3D transition (gametophore bud formation). Both have been extensively studied and genetically linked to central, conserved developmental regulators acting in both flowering plants and moss, like *APB* and *CLV*[28,30,41]. This observation supported a deep evolutionary conservation of the underlying developmental processes controlled by DEK1.

### Tracing misregulated molecular actors of pleiotropic DEK1 phenotype in the GRN

To understand the role of the predicted DEK1 targets in these conserved developmental processes and their role in the pleiotropic dek1 phenotype, we established a protocol to predict genes underlying specific phenotypic characteristics of the *dek1* mutant strains. The resulting Factorial Differential Gene Expression Network Enrichment Analysis (FDGENEA) method utilizes phenotypic traits (Supplementary Fig. S10a) for differential gene expression analysis. The DEGs displaying significantly altered transcript levels in association with one of 17 phenotypic factors (Supplementary Data S8), are then traced in the GRN to identify the predominantly affected subnetworks (Supplementary Fig. S10b, e).

Again, the observed, significant network associations demonstrated the importance of subnetworks X, II and V in the *dek1* phenotype (FDR < 0.01; Supplementary Fig. S10b). The resulting FDGENEA genes sets associated with each trait overlapped, but also displayed substantial portions of genes specifically misregulated in response to a single trait (Supplementary Fig. S10f). The phenotypic traits tested here clearly clustered into two classes enriched for either DEK1-controlled activator or repressor targets (Supplementary Fig. S10g), comprising 2048 indirect DEK1 targets.

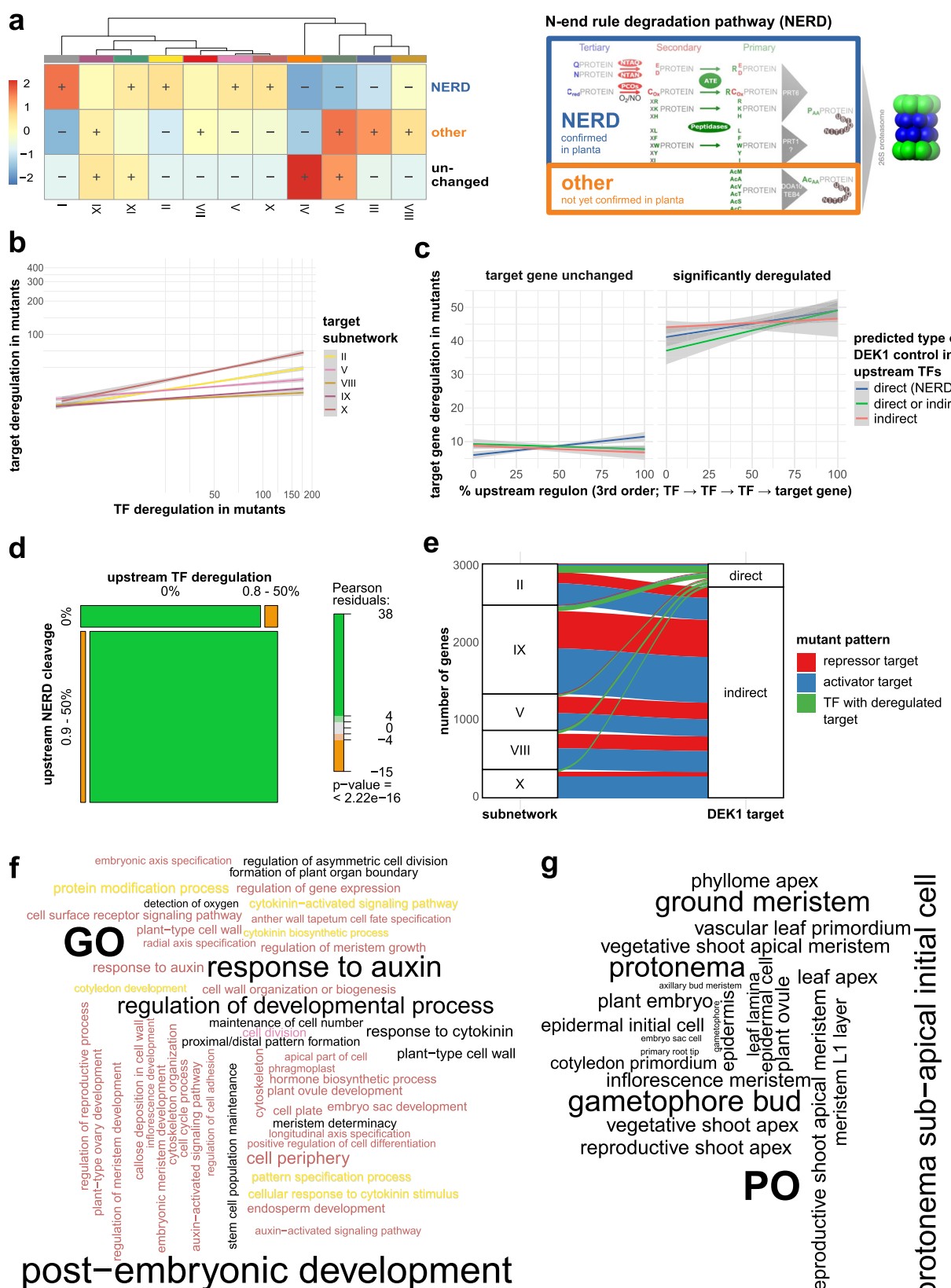

## Ectopic 3D stem cells linked to deregulation of bud cell-specific, DEK1-controlled genes

The earliest step of the 2D-to-3D transition that appears to be disrupted in the DEK1 mutants is the initiation of gametophore apical stem cells (buds) along the protonema filaments. Phenotypically, this is particularly pronounced in the number of buds per filament and the percentage of filaments with buds (Fig. 1c). While the *oex1* line forms fewer, the *Δdek1* and *dek1Δloop* lines develop significantly more buds per filament than the wild type (ANOVA with post hoc LSD, $p < 0.05$). This latter overbudding phenotype (Fig. 4a) is consistent with a disrupted control of gametophore

**Fig. 3 | Loss of DEK1 function causes global misregulation of the moss GRN in accordance with the pleiotropic phenotype of *dek1* mutants and is consistent with a post-translational role of plant calpain in the regulation of TF stability.** **a** Network enrichment analysis (NEAT) of the prevalence of three specific N-terminal amino acid signature types in predicted calpain cleavage sites of encoded moss proteins. Two of these signature types have been identified in mammals to activate the NERD pathway, resulting in ubiquitylation and subsequent degradation by the 26S proteasome (extended panel on the right adapted from ref. 34). While the first route has been confirmed to be active in the moss and other plants (NERD), the second route (other) via acetylation of N-terminal residues has not yet been demonstrated *in planta*. The third class represents proteins with no cleavage or sites with N-terminal residues that would not attract the NERD pathway (unchanged). The heatmap shows the ratio between observed and expected sizes of specific candidate gene sets encoding for proteins enriched for these types of cleavages among the identified subnetworks. Significant (FDR < 0.01) enrichment (+) or depletion (−) is shown. **b** Target gene misregulation in *dek1* mutants is positively correlated with the misregulation of direct upstream TFs. Linear relationship of target gene and TF misregulation in *DEK1* mutants in the five most affected subnetworks. Misregulation of both types of genes is again depicted as the cumulative effect size of the LRTs in each gene. Lines depict the result of generalized linear regression of the cumulative misregulation of TF genes (*x*-axis) and their target genes (*y*-axis) for each of the five subnetworks. Gray areas depict 95% confidence intervals. **c** Target gene misregulation shows a positive, linear correlation with the fraction of directly and indirectly DEK1 calpain-controlled upstream TFs. Linear regression analysis of the cumulative misregulation of target genes (*y*-axis; sum of Likelihood-ratio test (LRT) effect sizes) and the percentage of the upstream TFs for each gene where TFs are either directly NERD-targeted by DEK1 (blue line; i.e., classified as NERD-type cleavage, **a**), indirectly DEK1 targeted (orange line; i.e., significantly misregulated; contained in gene sets displayed in **b** and Fig. 1f) or either of the two types (green line) for unchanged (left plot) and significantly misregulated (right plot; FDR < 0.1)

target genes. Upstream regulons for each target gene in subnetworks II, V and X were evaluated up to third-order relationships. Gray areas depict 95% confidence intervals. **d** DEK1 calpain-dependent misregulation in the three subnetworks implementing the 2D-to-3D transition: misregulated genes in subnetworks II, V and X display significant enrichment of putative NERD-type calpain cleavages and misregulation of upstream TFs. Mosaic plot showing the relative proportions of significantly misregulated genes in *dek1* mutants depending on the binary status of their upstream regulon with respect to predicted levels of DEK1 control (*x*-axis: predicted NERD-type calpain cleavages i.e., direct DEK1 targets; *y*-axis indirect DEK1 targets). Binary status defines whether the regulon comprises TFs predicted as direct (*x*) or indirect (*y*) DEK1 targets (>0% of the TFs) or not (= 0% TFs). Boxes are colored based on Pearson residuals from a significant $\chi^2$ test of the cross-table comparing the proportions of both binary classes. **e** Alluvial plot depicting the distribution of the filtered, predicted direct and indirect DEK1 targets among the five predominantly controlled subnetworks. Color-coding of bands reflects directionality of misregulation patterns in the mutant lines (see Fig. 1f for details). Green bands represent unaffected upstream TFs predicted to control the significantly misregulated target genes. **f** Significantly enriched Gene Ontology (GO) terms associated with direct and indirect DEK1 target genes are overrepresented in processes related to observed DEK1 phenotypes in flowering plants and the moss. Word cloud depicts filtered, significantly enriched GO biological processes and cellular components (FDR < 0.1). Text color code depicts subnetwork identity (i.e., subnetworks II, V and X) of (indirect) target gene. Black text corresponds to overall enrichment among target genes. **g** Overrepresented tissue and cell type localizations consistent with *dek1* phenotypes and expression patterns. Based on enrichment analysis using Plant Ontology (PO) term annotations for moss genes or their flowering plant orthologs. Word cloud of selected plant anatomical entity PO terms displaying an overall enrichment among direct and indirect DEK1 target genes (FDR < 0.1).

---

initiation, leading to ectopic formation of 3D apical stem cells. It shows the largest unique set of deregulated genes in the FDGENEA of this group of traits (Fig. S10h and Supplementary Data S8) and is enriched for genes from subnetworks II, V, IX, X and XI (Figs. 4a, b and S10c, e).

The affected parts of the GRN (Figs. 4a and S10c, d) partitions into three groups of nodes. Two groups correspond to genes that are either positively associated with a high number of buds (overbudding-up; i.e., up-regulated in Δ*dek1* and *dek1Δloop*; right group in Fig. 4a and Supplementary Fig. S10c, d) or those that display a negative association (overbudding-down; i.e., down-regulated in Δ*dek1* and *dek1Δloop*, but up-regulated in WT, *oex1* and *dek1Δlg3*; left group in Fig. 4a and Supplementary Fig. S10c, d). The third group is composed of their direct upstream regulators without significant change in expression with respect to this phenotype (top group in Fig. 4a and Supplementary Fig. S10c, d). The clustering reveals also a trend in the type of connections between the first two groups that also harbor negative regulatory interactions (orange edge color; Fig. 4a and Supplementary Fig. S10c, d). Overall, while subnetwork V dominates the overbudding-down group, the overbudding-up assemblage is more diverse and consists of subnetworks X, IX and XI (Fig. 4b and Supplementary Fig. S10c). Subnetwork II is prominent in both groups. Negative inter-subnetwork links predominantly involve nodes between subnetwork V and either II or X. Subnetworks II and X as well as IX and XI seem to act in conjunction, i.e., share many positive edges (Supplementary Fig. S10c). These patterns are consistent with the global network structure discussed above (Fig. 2f) and the sequential transition between the encoded cell fates (Fig. 4f) from primary filament cells (V) redifferentiating to pluripotent side branch initials, that give rise to either secondary chloronemal (V) or caulonemal filaments (II) or gametophore buds (X).

The set of genes up-regulated in filaments displaying the overbudding phenotype is dominated by indirect and direct DEK1 targets from subnetworks II, IX and X (Fig. 4b). Down-regulated genes are either not targeted by DEK1 or encoded by subnetwork V or IX. There is a significantly larger proportion of DEK1-controlled regulatory interactions for overbudding associated genes in subnetworks II and X (Supplementary Data S9).

Subnetwork V displays more non-DEK1 controlled interactions, most being negatively associated with overbudding. While these regulatory interactions likely represent the side-branch initials redifferentiation into secondary chloronema (Fig. 4f; lower row), the overbudding up-regulated, predominantly DEK1-controlled interactions in subnetworks II and X likely encode the cell fate transitions required to establish the gametophore apical stem cells (buds).

Consistently, the set with positive association to overbudding (up in Fig. 4b) is enriched for DEK1-controlled activator targets from subnetwork II and X which have been previously identified to be specific to gametophore bud cells, while down-regulated genes from V are predominant in the protonemal tip cell transcriptome[23]. Overbudding-associated genes from subnetworks IX and XI are more likely to be found in both transcriptomes, hinting at their more ubiquitous expression profiles or housekeeping function. The bud-specific portion of overbudding-up DEK1 targets reveals 248 genes (Supplementary Data S10). The majority (73%) is encoded by subnetworks II (62; 25%) and X (118; 48%). Thus, the DEK1-controlled, overbudding up-regulated activator targets from subnetworks II and X represent prime suspects to harbor the developmental regulons acting in the cell fate transitions involved in gametophore apical stem cell initiation which is so pivotal to the 2D-to-3D transition.

## Overbudding up-regulated DEK1 targets form a regulon with known molecular actors of meristematic cell fate specification

Network analysis reveals that around 51% of the 901 overbudding-up DEK1 activator targets form an interconnected regulon (Supplementary Fig. S10i). Given the post-translational role of DEK1, a direct cleavage target will not be deregulated in the mutant context unless an upstream regulator is also a direct cleavage target. Thus, a regulon solely inferred based on the overbudding-up genes will be an under-prediction. Indeed, when we extend the regulatory context of these genes to include the up to five highest ranking DEK1-controlled upstream TFs, 100% of the overbudding-up genes are interconnected (Supplementary Fig. S10j).

Two regulatory circuits from subnetworks II and X control the majority of the overbudding-up components in the regulon (Supplementary

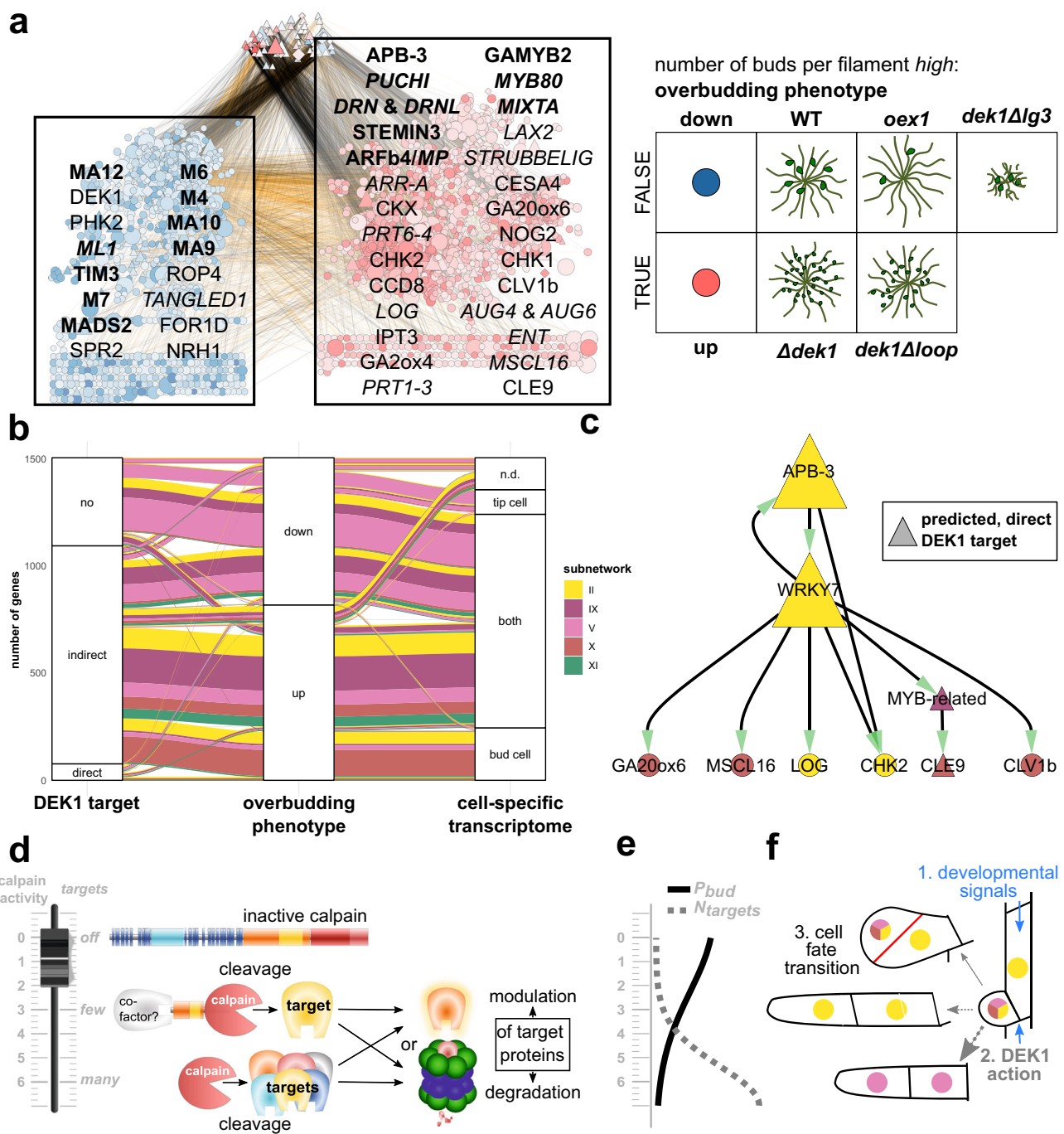

Fig. S10i). The first circuit is dominated by TFs from the AP2 superfamily in subnetwork II and comprises several well-characterized key players in meristematic cell fate regulation of flowering plants and *P. patens* (DRN, DRNL and PUCHI[42], STEMIN3[43], APB-3[30]). The second circuit is dominated by subnetwork X encoded, gibberellin-responsive MYB TFs that have been shown to orchestrate reproductive organ development as well as the production of extracellular hydrophobic barriers like the cuticle and sporopollenin of flowering plants and the moss (GAMYB2[31]; MYB80[44]; MIXTA[45]). While the two circuits are mostly insulated, some of the target genes overlap (17% of overbudding-up only- and 35% of the extended regulon; Fig. S10l, m). At the regulatory level, this insulation might be unidirectional in that one of the subnetwork X MIXTA orthologs is predicted to positively regulate a DRN and a DRNL ortholog in subnetwork II. This might represent a positive feedback mechanism.

Target genes of both circuits are involved in reorientation of the division plane, modulation of the cell wall, the cytoskeleton and the phragmoplast (e.g., Fig. 4a). Moreover, they encompass a notable accumulation of components required for generation, transduction and perception of local queues like mechanical stress (MSCL16[46]) and longer distance, gradient-forming, developmental signals like plant peptide hormones (CLV1b, CLE9[41]) or the phytohormones auxin, gibberellin, strigolactone and cytokinin. The latter is especially noteworthy. While the other three phytohormone pathways are represented by one or two components involving either transport (LAX2[47]), biosynthesis (GA2ox4[32]; CCD8[48]) or activation (GA20ox6[32]), in the case of cytokinin, all major aspects are covered. The regulon comprises several genes encoding biosynthesis (IPT3[49]), activation (LOG[50]), degradation (CKX[51]), transport (ENT, ABCG[52]) perception (CHK1, CHK2[51]) and transduction (ARR[53]) of cytokinins. In

**Fig. 4 | Tracing the overbudding mutant phenotype to deeply conserved, DEK1-guarded meristematic regulons controlling the 2D-to-3D transition. a** Factorial Differential Gene Expression Network Enrichment Analysis (FDGENEA) of the overbudding phenotype reveals enriched subnetworks and upstream regulators associated with high number of buds per filament that comprise key factors in plant meristematic and primordial cell fate control. Network plot of genes with significant association to overbudding (left [blue: overbudding = *FALSE*] and right node [red: overbudding = *TRUE*] groups in background of overlaid text boxes) and direct, upstream regulators without significant association (top node group). Foreground text boxes display exemplary, predicted DEK1 targets from the overbudding up regulated (right = red) and downregulated (left = blue) gene sets with experimental evidence in flowering plants or the moss. Bold font indicates TFs; italic font indicates predicted moss genes whose Arabidopsis orthologs have consistent experimental data connected to DEK1 phenotypes. Nodes are color-coded by the kind and strength of a gene's association with the overbudding trait (color intensity gradient relative to Log-fold-change in DGE analysis; up = positive = red; down = negative = blue). Node sizes relative to the cumulative, absolute misregulation fold-change of the respective gene and any predicted downstream target gene in the mutants. Node shapes: triangles, TFs; diamonds, transcription regulators; inverted triangles, miRNAs; circles, targets. Edge color and intensity: correlation coefficient of connected nodes in *dek1* RNA-seq data (black = positive; orange = negative). Panel at right depicts genotypes used and respective phenotypic character state of the overbudding trait (number of buds per filament *high*: FALSE ⇔ TRUE) in WT and *dek1* mutant lines. **b** Overbudding-upregulated DEK1 targets from subnetworks II and X are enriched in the previously identified bud cell transcriptome[23]. Alluvial diagram depicts the proportional distribution of subnetworks shared between three categorical sets: left to right, DEK1 target: predicted direct and indirect DEK1 targets; overbudding phenotype: genes with significant association with the overbudding phenotype (up ⇔ down); cell-specific transcriptome; n.d., not detected; specific to

protonemal tip cell; detected but no significant difference between both cell types; specific to gametophore bud cell. Band coloring is based on subnetworks. **c** Key components of the moss CLAVATA3-like peptide (CLE9) and receptor-like kinase (CLV1b) pathway are predicted to be downstream of an overbudding upregulated, DEK1-controlled regulon that comprises a 2D-to-3D master regulator (APB3) and integrates several developmental signals: gibberellin/kaurene (GA20ox6), cytokinin (LOG, CHK2), mechanical stress (MSCL16), peptide (CLE9). Network graph depicts the immediate regulatory context of CLV1b. Full regulatory context is shown in Supplementary Fig. S10i. Node sizes are relative to the overall local reaching centrality (fraction of downstream nodes in the global network). Node coloring based on subnetwork affiliation. Triangular nodes are predicted to represent direct cleavage targets of the DEK1 calpain. All predicted regulatory interactions are positive, i.e., show positive correlation in WT and mutants along the RNA-seq time course. **d**–**f** Proposed role of DEK1 as a fine-tunable, developmental switch gate-keeping cell fate transitions. Model illustrating the proposed relationship between the level of free calpain activity, the number of direct and indirect targets, and the developmental consequences in three panels with a shared *y*-axis (calpain activity). **d** Model describing three primary DEK1 calpain states (top to bottom): *off*: immobile, inactive calpain in full-length DEK1 protein in plasma membrane; *few*: mobile, constrained calpain, released by auto-catalytic cleavage at several possible locations in the Linker-LG3 domain (External File DEK1.jvp; level of calpain activity, localization, half-life and number/kind of targets might be dependent on co-factor interaction); *many*: mobile, unconstrained calpain, pure calpain released by auto-catalytic cleavage directly before or in the CysPc domain. **e** Proposed relationship between the number of DEK1 calpain targets ($N_{targets}$ = dashed curve) and the probability of a protonemal cell gaining the bud initial cell fate, i.e., gametophore apical stem cell ($P_{bud}$ = solid curve). **f** Schematic drawing of the cellular fate transitions affected in the overbudding phenotype in three steps.

---

addition, the MYB circuit of the regulon also is predicted to induce cytokinin-responsive genes like NO GAMETOPHORES 2 (NOG2[54]), whose loss-of-function mutant similar to DEK1 also displays an over-budding phenotype. Consistently, the exogenous application of cytokinin[51] and cytokinin-overproducing mutants[55] also result in an overbudding phenotype. These findings clearly demonstrate the conservation of hormonal control in stem cell initiation and cell fate specification in land plants[56].

As mentioned above, the regulon also comprises essential components of the CLAVATA (CLV) peptide and receptor-like kinase pathway that has been shown to control cell fates and division planes of land plant apical stem cells[41,57,58] via CLV3/EMBRYO SURROUNDING REGION-Related (CLE[59]) peptide hormones which are perceived and transmitted to downstream signaling cascades via CLV1-type receptor-like kinases[60]. Several studies in both Arabidopsis[61] and Physcomitrella[41,54,62] already have identified parallels in mutant phenotypes and expression patterns and have proposed models locating the CLV signaling pathway somewhere downstream of DEK1 and the moss APBs. The predicted overbudding-up regulon identified here, now provides us with a robust explanation for these connections. Our predictions indicate that CLV1b and CLE9 are an integral part of the regulon that is downstream of both of the above DEK1-controlled circuits (Fig. S10i, k), in particular downstream of APB-3 (Fig. 4c). APB-3 is predicted to coordinate its control over these genes with a calmodulin-binding WRKY group II transcription factor (WRK7[63]), that could act to integrate a possible Ca$^{2+}$ signal emerging in response to the swelling of the gametophore initial cell[64]. Furthermore, our calpain cleavage predictions indicate cleavage sites that would allow the maturation of CLE peptides from their respective preproteins encoded by the *P. patens* genome[41] (External File PpCLEs.ccd.all). This could represent another potential feedback layer of the regulon in that CLEs are both positively (maturation/activation) and negatively (indirect activator targets) controlled by DEK1. The second order regulatory context of CLV1b (Figs. 4c and S10k) suggests that all positional and developmental queues discussed above are co-regulated in one DEK1-controlled CLAVATA regulon. Consistent with the findings from two recent studies[65,66], our data suggests that this DEK1-controlled, cytokinin-mediated pathway governs stem-cell homeostasis acting separately from the

cytokinin-independent pathway involving the RECEPTOR-LIKE PROTEIN KINASE2 (RPK2; subnetwork VIII).

The overbudding-down part of the GRN is controlled predominantly by MADS box TFs, contains several correctly predicted, negative regulators of bud and gametophore formation (e.g., DEK1[11] PHK2[67]) and is enriched for cytoskeletal components involved with polarized tip growth of protonemal filaments (e.g., FOR1D[68] SPR2[69]). Plant Rho GTPases (ROP) are key regulators of cellular polarization and are involved in several symmetry breaking mechanisms[70,71]. Activated ROP binds effector proteins e.g., to initiate remodeling of the cell wall (96) or the cytoskeleton[71]. Sometimes they act as transducers for receptor-like kinases[72]. The *P. patens* ROP4 is localized at the tip of a growing protonema filament and relocalizes prior to protonemal branching to the future site of side branch formation[73]. ROP4 is predicted to be an overbudding-down DEK1 repressor target. Rho GTPase-dependent signaling by ROPs is tightly controlled at the protein level[70]. ROPs are activated by RhoGEFs, while RhoGDIs and RhoGAPs provide independent means of ROP inactivation. Our analysis detected a representative of both ROP-regulator types as overbudding-associated, indirect, DEK1-targets with opposing regulatory patterns (DEK1 activator target, overbudding-up, part of the CLAVATA regulon: ROP-GEF, Pp3c10_9910; DEK1 repressor targets, overbudding-down: RhoGAP, Pp3c3_5940 and RhoGDI, Pp3c10_19650). This observation is consistent with their proposed antagonistic role in controlling ROP signaling. It provides a compelling example of how DEK1 might post-translationally control asymmetric and other types of formative cell division by remodeling of cell walls and the cytoskeleton.

## Discussion

Here, we traced the misregulation profiles of null *dek1* mutants and overexpressor lines along the GRN of the model plant *P. patens*, identifying at least 3679 consistently misregulated genes whose expression is controlled by 531 upstream TFs containing destabilizing calpain cleavage sites. We propose that these TFs are direct targets of DEK1, which thus acts as an indirect regulator of genes farther downstream. Individual master regulators and downstream TFs, and many of the target effector genes, have been experimentally linked to specific *dek1* mutant phenotypes in *P. patens* and in

flowering plants. We conclude that DEK1 exerts a dual role as a modulatory and destabilizing protease acting on both the physical and regulatory layers of cell fate transitions, thereby indirectly controlling the expression levels of many genes.

This post-translational role in gene regulation and the predicted list of DEK1 targets provide a consistent explanation for the essentiality of this calpain and for the pleiotropy and broad effects observed in *dek1* mutants. We based our predictions on the expression profiles of mutants, together with the inferred GRNs of *P. patens*. The breadth of these filtered cleavage sites, especially in TFs and other gene regulators, is consistent with the observed broad transcriptional, functional and phenotypic responses observed in *P. patens*.

Individual examples like ROP signaling or the above-described CLA-VATA regulon may help to bridge the gap between the well-established image of DEK1 as a developmental regulator that is affecting cell fates and division plane reorientation, and the role as a post-translational regulator proposed here. Our analyses suggest the existence of deeply conserved, orthologous, phytohormone-guided regulons governing land plant meristems and stem cells that are post-translationally controlled by DEK1, which acts as a fine-tunable switch to implement or guard the transitions between cellular identities. The high levels of conservation in the underlying regulatory network highlights the utility of *P. patens* for elucidating embryophyte development and stem cell regulation. We showed here that DEK1 is a negative regulator of cell fate transitions, specifically during the 2D-to-3D transition.

In our model, DEK1 integrates multiple developmental signals (phytohormones, peptides, light, mechanical stress) and acts as a gatekeeper in the transition between distinct cellular fates (Fig. 4d–f). The transcriptional profiling of the four distinct *dek1* mutant lines enabled us to monitor three extreme points in the distribution of calpain activity: the number of direct and indirect targets ($N_{targets}$; off = $\Delta dek1$, $dek1\Delta loop$; few = $dek1\Delta lg3$; many = $oex1$; Fig. 4d, e). With their intermediate phenotypes and misregulation patterns, the two partial deletion lines indicate how the distinct functional regions of DEK1 might be involved in fine-tuning free calpain activity. In our model for gametophore bud formation, the level of calpain activity is proportional to the probability ($P_{bud}$; Fig. 4e) of a side branch initial developing into a gametophore initial cell (Fig. 4f).

The immobile, inactive, full-length DEK1 protein (off; Fig. 4d) resides at the plasma membrane[31] and can potentially be phosphorylated at several sites[74,75], probably resulting in conformational changes and (de)activation. While animal calpain activity depends on $Ca^{2+}$ binding[76], it is currently unclear to what extent $Ca^{2+}$ activation is required for the DEK1 calpain's CysPc-C2L protease domain[77,78]. The autocatalytic activity of DEK1 (External File DEK1.jvp) likely results in a short half-life of the mobile, unconstrained calpain that can target many proteins, potentially acting as a reset switch affecting turnover of the entire or large parts of the cellular protein complement (Fig. 4d). This is presumably the highest level of calpain activity with a short half-life of individual calpain molecules.

However, not all potential cleavage targets bear destabilizing N-terminal residues targeting a protein for proteasomal degradation via the NERD pathway (Fig. 3a). Depending on the amino acid signature of the new N terminus, the resulting polypeptide may be either NERD-directed or stable and may represent the activated or mature form of the protein or peptide (e.g., CLEs), which may also hold true for DEK1 itself. Our data suggest that at least three stable DEK1 variants potentially arise by autocatalytic cleavage in the Linker domain (External File DEK1.jvp). These are similar to the sizes of experimentally confirmed forms in *Arabidopsis thaliana*[61]. The varying N-terminal regions resulting from such cleavages might lead to different half-lives or modify calpain's specificity or target range (Fig. 4d).

We also found components of the NERD pathway (e.g., orthologs to crucial N-recognins PRT1 and PRT6; Fig. 3a)[41] among the indirect DEK1 targets. These potentially represent yet another regulatory layer, allowing to switch off degradation or fine-tune protein stability and balance post-cleavage protein fates toward the modulator activity i.e., activation or maturation (Fig. 4d).

Calpain research has largely focused on calpains' roles as non-processive, modulatory proteases[18]. Much less attention has been paid to their destabilizing characteristics, observable in the coactivation with the ubiquitin-proteasome system and the generation of short-lived substrates for the NERD pathway[19]. Importantly, many experimentally characterized calpain targets, especially those with confirmed NERD degrons[19], are involved in transcriptional or other gene regulation. The functional implications of this have so far been under-investigated.

It has been difficult to align the observed directionality (activation vs. inactivation of biological functions), effects, pleiotropy and severity of *dek1* phenotypes with DEK1's role as a sole modulator protease. Our observations identify DEK1 as an upstream component of the ubiquitin-proteasome system that directs proteins via the NERD pathway. DEK1 cleavage of activating/repressing TFs can inhibit/activate the expression of all downstream target genes and thus indirectly regulate gene expression, largely explaining the substantial changes in gene expression observed in *dek1* mutants. Nevertheless, for some targets, DEK1 may act as a non-processive and modulatory protease, like other calpains. Our predictions do hint at the importance of DEK1 in protein maturation and activation (CLEs). Thus, we propose a duality of outcomes for proteolysis by calpains (Fig. 4d). Our data in *P. patens* indicate that the final outcome of cleavage is usually degradation by the proteasome.

Calpains participate in a spectrum of biological processes and are controlled at multiple levels[16]. The proposed dual role for DEK1 as a modulatory and destabilizing protease that modulates a fraction of protein functions, while directing most detrimental cleavage fragments toward the NERD pathway for degradation, provides the most parsimonious explanation for our observations. The gene-regulatory consequences and effects of NERD pathway control over calpain-targeted TFs allowed the establishment of this route as a regulatory mechanism in the form of a post-translational gatekeeper of cell fates. The fact that *DEK1* is a single-copy gene in most land plants argues for a crucial and dosage-sensitive role of this plant calpain[8,11,15].

Systematic, large-scale analysis or discovery of calpain targets has been hindered by their limited target specificity, their involvement in a broad spectrum of biological processes and the complexity of their regulatory mechanisms. Considering the confirmed calpain-targeted human gene regulators and reports of gene misregulation in metazoan calpain mutants and human pathologies, metazoan calpains, too, might have gene-regulatory roles. The developed FDGENEA method can also prove to be invaluable to other sorts of genotype-phenotype mappings. Our approach of tracing calpain mutant or misregulation profiles in GRNs to identify indirect and direct targets might help elucidate this under-explored aspect of calpain biology more broadly. The resulting genome-wide, unbiased target candidate gene lists are valuable starting points for mechanistic exploration of this enigmatic major proteolytic system with important regulatory and developmental implications in all eukaryotes.

## Methods
### Plant materials and growth conditions
*Physcomitrium* (*Physcomitrella*) *patens* Gransden WT strain and four mutants, $\Delta dek1$[10], $dek1\Delta loop$[11], $dek1\Delta lg3$[14] and *oex1* (this work), were used. Protonemata were maintained on minimal medium supplemented with 920 mg l$^{-1}$ ammonium tartrate (BCDA medium) under a 16-h light (70–80 mmol m$^{-2}$ s$^{-1}$)/8-h dark photoperiod at 25 °C. Cultures for phenotypic characterization and RNA extraction were grown under the same conditions on minimal BCD medium with no ammonium tartrate added[10].

### Generation of the DEK1 Linker-Calpain overexpressing strain *oex1*
The cDNA encoding the Linker-Calpain domains was PCR amplified with primers P1 and P2 (Supplementary Data S12)[11]. The PCR amplicon was cloned into the pCR8/GW/TOPO TA vector (Invitrogen), and mobilized into the pTHUBI Gateway vector[79] using LR Clonase (Invitrogen). The vector allows expression during the entire moss life cycle, and its targeting to

the *108* locus does not induce phenotypic changes[80]. For transformation, the targeting fragment was amplified by PCR using the primers P3 and P4 designed at each end of the targeting sequence.

Transformation of WT *P. patens* was performed via polyethylene glycol (PEG)-mediated transformation of protoplasts[10]; the Linker-Calpain overexpressing strain *oex1* was selected for further analysis. In parallel the *oex1* moss went through a cycle of sexual reproduction[36] and displayed normal sporophyte development. Subsequent spore germination and gametophyte development were consistent with the observed phenotype of the original *oex1* transformant (Fig. 1). PCR genotyping showed 5' targeting of the construct at the *108* neutral locus[81]. A Southern blot[10] also indicated that *oex1* harbors multiple copies of the construct at the targeted locus (Supplementary Figs. S11 and S13). Genomic DNA for Southern blot analysis was extracted using the NucleonTM PhytoPureTM Genomic DNA Extraction Kit (GE Healthcare). Southern blotting was performed using 1 µg genomic DNA per digestion. Probes were labeled with digoxygenin (DIG; Roche, Indianapolis, USA). DNA from the pTHUBI Gateway vector[36] was used as template for PCR amplification of the TS and hygromycin-resistance probes with primers 108_5fw and 108_5rev (5′ TS probe); 108_3fw and 108_3rev (3′ TS probe); HRC-fwd and HRC-rev (HRC probe; Supplementary Data S12). Immunoblotting using the anti-PpDEK1 specific antibody anti-CysPc-C2L (GenScript, produced in rabbit, epitope sequence WSRPEEVL-REQGQDC) confirmed the accumulation of the Linker-Calpain protein (Supplementary Figs. S11 and S13). For protein extraction, tissue from 12-day-old cultures was homogenized in liquid nitrogen and 300 µg of powder was resuspended in 600 µl of extraction buffer (0.43% [w/v] DTT, 6% [w/v] sucrose, 0.3% [w/v] $Na_2CO_3$, 0.5% [w/v] SDS, 1.0 mM EDTA, Roche cOmplete Protease Inhibitor Cocktail).

Samples were incubated at 70℃ for 15 min and centrifuged at 2000 rpm for 10 min. Proteins were separated on 4–15% Mini-PROTEAN TGX Gels (Bio-Rad) and transferred onto nitrocellulose membranes using Trans-Blot Turbo Transfer Packs (Bio-Rad). Membranes were incubated with anti-CysPc-C2L primary antibody diluted 1:500 in Tris-buffered saline + Tween-20 (TBST) containing 5% (w/v) skimmed milk. A goat anti-rabbit secondary antibody (IgG [H + L]) conjugated to HRP (Bio-Rad) was used and signal was detected using Clarity Max Western ECL Substrate (Bio-Rad) according to the manufacturer's protocol.

### DEK1 protein domain structure
Positions of the 23 transmembrane helices in Fig. 1a (DEK1 MEM, dark blue) inferred by MEMSAT3[82]. Positioning of the remaining domains (DEK1 Linker, DEK1 Calpain) is based on previous phylogenetic analyses[14].

### Statistics and reproducibility
Statistics and other data analyses were implemented as described below using custom R or Python scripts and Jupyter notebooks (see "Data availability" statement below). Where applicable, a common random seed number was utilized for the presented final description and visualizations and is provided with the respective source code or configuration file. Where applicable reproducibility was assessed by testing different seed numbers. Wet lab experiments were carried out at least in triplicates. Individual number of replicates, sample sizes and description of reproducibility is provided in the respective method descriptions and figure legends. Analysis of variance (ANOVA) and least significant difference (LSD) test were performed in multiple sample comparisons presented in Fig. 1c. All data were provided as part of the Supplementary Materials and Data or deposited in FAIR data repositories.

### Time series analysis of *P. patens* juvenile gametophyte development
For comparison of juvenile gametophytic development in the WT and *dek1* mutants (Fig. 1b and Supplementary Figs. S9 and S10a), tissue from 1-week-old protonemata cultures was homogenized in sterile water and inoculated onto minimal medium (BCD) overlaid with cellophane. Material for RNA extraction was harvested after 3, 5, 9, 12 or 14 days of growth, always at the same time of day. The samples were frozen in liquid nitrogen and stored at −80 ℃ until processing. Three starting cultures for each strain were used to initiate parallel cultures (biological replicates) used for RNA extraction. Phenotypic characterization of the plant material was performed using light microscopy and image analysis using ImageJ software.

### RNA extraction, RNA quality assessment, RNA sequencing of *dek1* mutants
Total RNA was extracted from frozen material using the RNeasy lipid tissue mini kit (Qiagen) with few modifications. Briefly, the frozen tissue was thoroughly homogenized using a tissue lyser with pre-frozen blocks. Approximately 120 µg of powdered tissue was lysed in 1 ml of QIAzol lysis reagent. Then, 200 µl of chloroform was added and the mixture was centrifuged at 4 ℃. The aqueous phase was collected, 1.5 volumes of 100% ethanol was added to it, and the mixture was vortexed. After binding of the RNA to a RNeasy mini spin column, on-column DNase I digestion was performed to remove genomic DNA. The column was washed with RPE buffer (Qiagen) and air-dried, and the RNA was eluted in 45 µl ribonuclease-free water. The concentration of total RNA was measured and RNA integrity was assessed using an Agilent 2100 Bioanalyzer (DE54704553; Agilent Technologies) with an RNA 6000 LabChip kit. The RNA samples were stored at −80 ℃ until being sent for sequencing. Strand-specific TruSeq™ RNAseq library construction of 74 libraries and sequencing using a HiSeq2500 instrument (Illumina) as 125-bp paired-end reads were performed.

### RNA-seq data collection, read quality analysis and mapping
In total, 299 publicly available RNA-seq libraries for *P. patens* were downloaded from EMBL ENA service. With the 74 RNA-seq libraries produced in this study, 373 libraries were analyzed in total.

Raw data were quality-checked using FastQ[83] and trimmed to remove adapter contamination and reads of poor quality using Trimmomatic[84].

### Non-redundant gene annotation, phylogenomics framework, regulator classification, improved ontology annotation and updated gene names
For optimal gene-level RNASeq quantification results, a non-redundant transcript representation of the v3.3 cosmoss genome annotation of *P. patens*[22] was generated. To this end, GFF3 transcript features of protein-coding and non-protein-coding genes were exported using gffread[85] to FASA and independently clustered at 100% sequence identity using CD-HIT[86]. The v3.3 genome annotation contains genes encoding both mRNAs and ncRNAs. As these two transcript types might represent opposite regulatory outcomes (e.g., an antisense transcript to a protein-coding mRNA), they were analyzed independently. The original v3.3 gene ids were extended by adding the primary tag of the transcript feature (i.e., mRNA vs. ncRNA, tRNA, miRNA or rRNA). Resulting transcripts were traced to genes using the original GFF3 parent-child relationships.

Gene families were defined in an automated, phylogenomics approach incorporating protein sequences from 69 Viridiplantae genomes (Supplementary Data S13) using OrthoFinder[87]. Homologous relationships among gene family members were analyzed by species tree reconciliation of gene trees to infer orthologs, inparalogs and outparalogs. Transcription factors, transcriptional regulators and other transcription associated proteins were inferred based on gene family membership and classification of domain architectures using the TAPScan rule set[88].

Inferred orthologous relationships were used to transfer automatic and experimentally validated annotations from orthologous genes. Gene Ontology[89] and Plant Ontology[90] term annotations were obtained and pooled from Gene Ontology (http://geneontology.org/gene-associations), TAIR (https://www.arabidopsis.org), and Gramene (ftp://ftp.gramene.org/pub/-gramene/release52/data/ontology) resources. Gene identifiers were mapped to public resources using the UniProtKB mapping table (ftp://ftp.uniprot.org/pub/databases/uniprot/current_release/knowledgebase/idmapping).

The pfam2GO mapping table available from the Gene Ontology resource (http://geneontology.org/external2go/pfam2go) was also employed to transfer GO terms based on the inferred domain architectures. The source evidence classes of the annotated, orthologous genes were translated into target evidence codes of *P. patens* genes as follows: (1) automatic annotations: IEA (Inferred by Electronic Annotation) (2) experimental and reviewed computational analyses (for full list of evidence codes in these categories see http://www.geneontology.org/page/guide-go-evidence-codes): e.g., EXP (Inferred from Experiment) and e.g., RCA (Reviewed Computational Analysis) and ISO (Inferred by Sequence Orthology) (3) pfam2GO: ISM (Inferred from Sequence Model). Subcellular localization predictions using YLOC[91], TMHMM[92] and MEMSAT3[82] that were translated into GO subcellular localization terms. Existing cosmoss *P. patens* v1.6 GO and PO ontology annotations were integrated[90,93] Altogether, extended annotation comprising 336 K GO terms and 877 K PO terms was used for the various ontology term enrichment analyses.

Gene names were transferred from the community-curated cosmoss legacy annotations and updated throughout the project to incorporate names from published moss and orthologous plant genes relevant to the study. Final gene names, description lines, regulator and superfamily classifications are provided as part of the External Files (listed in S13; genome_annotation/Physcomitrium_patens.names_and_regulators.tsv).

## Differential gene expression (DGE) analyses
Preprocessing, filtering and preliminary analysis of all DGE analyses conducted in this study were implemented in the Jupyter Notebook dge_analysis/ SetAnalysis.factors.ipynb. Analysis of DEGs including definition of the repressor and activator targets sets was conducted using the UpSetR R package[94] (R Jupyter notebook dge_analysis/SetAnalysis4Paper.ipynb).

## Ontology term enrichment in deregulated genes
Ontology term enrichment analyses for the distinct sets of DEGs obtained from the pairwise comparisons of wild type and mutant genotypes (Supplementary Fig. S1 and Supplementary Data S2), were carried out using the Snakemake workflow ontology_enrichment_workflow that builds on the Ontologizer software to test multiple sets in parallel for enrichment of terms in any OBO formatted ontology. Percentages of deregulated ontology terms for each mutant genotype were calculated and drawn in the Jupyter notebook ontology_enrichment_workflow/PercentDeregulated.ipynb (Supplementary Fig. S1).

## Quantitative analysis of gametophore meristematic bud formation in dek1 mutants
The frequency of gametophore apical stem cell initiation (Fig. 1c) was expressed as the number of buds formed per 15-cell-long filament and as the percentage of filaments forming buds. One hundred filaments from each strain were analyzed.

## RNA-seq analysis and expression matrix
Paired-end reads were aligned to the set of 80,244 *P. patens* unique transcripts and quantified with Kallisto applying 100 bootstrap replicates using the Snakemake workflow workflow_kallisto. Bootstrapped, individual transcript abundances obtained from kallisto were used for downstream analysis of differential gene expression (see below). To generate the input expression matrix for gene-regulatory network (GRN) analysis, gene-level transcripts per million (TPM) values were calculated using the R package tximport and then normalized using the variance-stabilizing transformation (VST) implemented in the DESeq2 R package (implemented in the Jupyter notebook grn_analysis/getGeneMatrix.ipynb).

## Pairwise, differential time series gene expression analysis of dek1 mutants and the WT along the developmental time course
Based on the bootstrapped kallisto transcript abundances, we performed pairwise, differential time-series gene expression (DGE) analysis of the *dek1* mutants and the WT using the response error linear modeling implemented

in the sleuth R package (Jupyter notebooks in folder dge_analysis/ TimeSeriesAnalysis.*_vs_*.ipynb).

To identify differentially expressed genes (DEGs) between genotypes, pairwise comparisons were undertaken. DEGs were inferred using the LRT false discovery rates (FDR; *qval* Supplementary Data S1*)* at 10% and 1% confidence. Directionality of differential expression (upregulation or downregulation; Fig. 1d, f) was defined based on the *b*-value obtained from Wald's test.

To identify DEGs during the WT developmental time course, we selected only the WT samples and performed likelihood ratio and Wald's testing comparing the B-spline time-series matrix as described above (Jupyter notebook dge_analysis/ TimeSeriesAnalysis.WT.ipynb). FDR cutoff values were chosen accordingly.

To identify DEGs between the early (3–5 days) and the late (9–14 days) phase of WT development (Fig. 2a), we selected only the WT samples and performed likelihood ratio and Wald's testing comparing the two phases in the full model versus the null model (Jupyter notebook dge_analysis/ TimeSeriesAnalysis.WT.early_vs_late.ipynb). FDR cutoff values were chosen accordingly.

To define patterns or profiles of misregulation of repressor and activator targets, DEGs were filtered using the R Jupyter notebooks dge_analysis/profile_phases/ identify_profiles.ipynb and dge_analysis/ profile_phases/getRelaxed.ipynb. For profile 1, we selected genes significantly upregulated in WT compared to *oex1*, downregulated in WT compared to *Δdek1* and downregulated in *oex1* compared to *Δdek1*. For profile 2, we selected genes significantly downregulated in WT compared to *oex1*, upregulated in WT compared to *Δdek1* and upregulated in *oex1* compared to *Δdek1*. For an additional description of the different phases along the time series, each profile was clustered using *k*-means clustering into three clusters. These clusters were manually interpreted and translated into phase descriptions.

We assessed the outcome of our time-series DGE analysis strategy with two conventional pairwise DGE analysis approaches (Fig. S12). While we observed a large consistency of results (e.g., 76% of the DEGs identified by sleuth also identified with edgeR in comparing up-regulated genes in the DEK1 null mutant), the sleuth DEG sets had better consistency with previous data11 as well as a more intuitive and condensed representation of the resulting DEG sets in our approach with kallisto/sleuth.

## Prediction and characterization of gene-regulatory interactions and subnetwork inference
Regulatory interactions were predicted in the genome-wide VST-transformed expression matrix based on 1736 regulators using the random forest predictor in GENIE3[26] (R Jupyter notebook grn_analysis/GENIE3.ipynb). A set of 992 TF genes, 413 transcriptional regulator (TR) genes, 79 putative transcription-associated (PT) genes, 275 microRNAs and *DEK1* were specified as candidate regulators.

The overall directionality of regulatory interactions was determined by Pearson's correlation coefficients between the expression levels of the regulator and its target gene along the developmental time course in the WT and *dek1* mutant samples as well as globally using all columns of the matrix (Python Jupyter notebook grn_analysis/GetCorrelation.ipynb).

Community detection was carried out using the Parallel Louvain Method implemented in the NetworKit Python package based on the top 10 regulatory interactions of each target gene[95] (Python Jupyter notebook grn_analysis/GetCommunities.ipynb).

To characterize the connectivity of nodes and rank the nodes, several centrality measures were calculated, which were implemented using the NetworKit Python package (Python Jupyter notebook grn_analysis/Get-Communities.ipynb). These values leading by the local reaching centrality were used to sort and rank the nodes to establish a regulator hierarchy.

The barplot summarizing the subnetwork structure (Supplementary Fig. S2a) of the *P. patens* GRN was drawn with the Jupyter notebook grn_analysis/PlotSubnetworkMisregulation Patterns.ipynb.

### Differential gene expression (DGE) analysis of developmental stages in the *P. patens* Gene Atlas

Based on the bootstrapped kallisto transcript abundances of the *P. patens* Gene Atlas data set[24] (dge_gene_atlas/subset.full_metadata.txt), we carried out DGE analysis for each of the five covered developmental stages (spores, protonema, gametophores, green sporophytes, brown sporophytes) using the sleuth R package[25] as described above for the developmental time course. In this case, the full model compared samples for each developmental stage against all other samples. Analyses were implemented in the Jupyter notebooks dge_gene_atlas/DGE.gene_atlas.*.ipynb.

### Functional characterization of the subnetworks

We utilized the developmental stage samples included in the *P. patens* Gene Atlas[24] as well as Plant Ontology (PO) and Gene Ontology (GO) annotations for functional characterization of the subnetworks. Both approaches independently considered the network's structure to assess over-representation of functional concepts among the genes in the network. Combined with the manually curated set of experimentally/genetically characterized moss genes, the two analyses provided the basis for the assignment of subnetworks to tissue and cell types (Fig. 2c, d) and to sub-cellular compartments (Fig. 2e).

Network enrichment analysis for the Gene Atlas developmental stage DGE sets defined at FDR < 0.1 (Supplementary Fig. S3a) was carried out using the NEAT R package[27] in the Jupyter notebook grn_analysis/NEAT.DGE.ipynb as described above for the DEK1 DGE sets.

The ontology analysis comprised a multi-step procedure relying on a machine learning approach to identify the most specific and characteristic terms for the genes encoded in each subnetwork. The final set of most characteristic PO and GO terms for each subnetwork (Fig. S3e–i) comprises the ontology terms that were most informative to classify the top20 master regulators from each subnetwork according to their targets' functional composition.

Primary ontology term enrichment analysis for the subnetworks was carried out using the ontology_enrichment_workflow as described above for the DGE sets. Total number of enriched terms at FDR < 0.1 for each partition of GO and PO in this primary analysis (Supplementary Fig. S3b) was analyzed and plotted using the R Jupyter notebook grn_analysis/Study Enrichments.ipynb.

Specificity of the primary analysis was analyzed via set analysis of the enriched GO biological process concepts (Supplementary Fig. S3c) using the UpsetR package (R Jupyter notebook grn_analysis/StudyEnrichments.ipynb).

The sources of ontology term annotations are manifold and differ in quality, resolution and intention. Genes can be experimentally connected to multiple processes while their direct functional involvement is limited to only some of them. Primary enrichment analysis does not consider the relationships between genes. As functionally related genes tend to be co-regulated, GRNs provide an additional layer to mine functional relationships. Even more so in our case, where we are interested in identifying the predominant biological processes and anatomical structures etc. encoded by each subnetwork. Thus, the subsequent steps were directed to integrate the information from the directed graph structures of the predicted GRN of *P. patens*.

Directed network enrichment analysis tests were carried out for each enriched ontology concept among the subnetworks using the NEAT R package[27] filtering terms at FDR < 0.01 in Jupyter notebook grn_analysis/NEAT_enriched_terms.ipynb.

In a next step, we constructed a regulator matrix where the 2084 columns contain for each NEAT enriched ontology term the annotated downstream gene frequencies for 1667 regulators (R Jupyter notebook grn_analysis/GetRegulatorMatrix.ipynb).

This matrix was used for a machine learning approach to identify distinctive ontology concepts for the top 20 regulators of each subnetwork using Random Forest classification implemented in the randomForest R package[96] (Jupyter R notebook grn_analysis/enriched_terms_selection_by_Random

Forest_variableImportance.ipynb). For preprocessing, the regulator matrix was further filtered to discard non-plant GO concepts as well as terms from either PO or GO that did represent ≥10% of the respective ontology's annotated gene space in at least one of the subnetworks. Individual regulators' ontology term gene frequencies of the remaining 379 columns were scaled using the overall number of genes annotated with each term and the terms information content. The rows of the matrix were filtered selecting only the top 20 master regulators for each subnetwork using the centrality rank criterion (220 regulators in total). The resulting, filtered matrix was used to train a Random Forest classifier with 100,000 trees recording variable importance i.e., each ontology term's importance to discriminate a subnetworks regulator from those of other subnetworks. Multidimensional scaling (MDS) plot of the classifiers' proximity matrix was carried out to analyze the conceptual similarity of the subnetworks (Supplementary Fig. S3d). The top 5 most specific terms to describe targets in subnetworks II, V, VIII, IX and X were plotted as word cloud representations (Supplementary Fig. S3e–i). We selected and ranked terms for each subnetwork demanding variable importance >0 using the decrease in node impurity based on the Gini index implemented by the R/randomForest package.

To identify and rank the five subnetworks contributions to the seven major subcellular compartments depicted in Fig. 2e, we semi-automatically screened, sorted and ranked the distinctive terms by their gene frequencies in the subnetworks using the bash shell (ontology_enrichment/syntax.get_DEK1_Fig2_numbers).

### Identification of the five DEK1-controlled subnetworks

The identification of predominantly DEK1-controlled subnetworks encoding the 2D-to-3D transition (Fig. 2a and Supplementary Fig. S2b) was carried out by tracing the overrepresentation and under-representation of the relevant DGE sets via a Network Enrichment Analysis Test (NEAT[27]) implemented in the R Jupyter notebook grn_analysis/PlotSubnetworkMisregulationPatterns.ipynb. *p* values were adjusted for multiple testing using the Benjamini-Hochberg method and filtered at 99% confidence.

The overall network structure of the putative indirect DEK1 targets (Fig. 2b) was analyzed and drawn in Cytoscape applying the AutoAnnotate[97] app.

### Identification of the major inter-connections between the five DEK1-controlled subnetworks

To identify the major regulatory interactions between subnetworks (Fig. 2f), we analyzed the cross-sectional distribution of inter-subnetwork connections considering the predicted directionality based on the Pearson correlation coefficient of expression profiles between a regulator and a predicted target from another subnetwork.

In the Jupyter notebook grn_analysis/PlotSubnetworkDeregulationPatterns.ipynb we utilized stacked barcharts in polar coordinate plots, a mosaic plot depicting the distribution of Pearson residuals indicating significant over- or under-representation of inter-connecting edges obtained from a significant $X^2$ test (*p* value < 2.22e−16; Supplementary Fig. S4) and cross-tabulation via $X^2$, Fisher and McNemar tests implemented in the gmodels R package[98] to assess the distribution of inter-subnetwork edges. The graph of inter-subnetwork connections shown in Fig. 2f depicts the major, significantly enriched inter-subnetwork connections with Pearson residuals >4 ($X^2$ test).

### Subgraph analysis—regulatory hierarchies, regulons and regulatory contexts

To analyze and visualize regulatory hierarchies, regulons and compare regulatory contexts (Supplementary Figs. S5a, b and S10i, k), we utilized the Cytoscape and the k-shortest paths algorithm with additive edge weights implemented in the PathLinker app[99]. We used the edge weights computed by GENIE3[26] and usually explored several k's to optimize resolution.

## Prediction of calpain cleavage sites and classification of potential for NERD targeting

Calpain cleavage sites were predicted for all predicted protein isoforms encoded by the *P. patens* V3.3 genome using GPS-CCD[35]. The results were classified using regular expressions capturing the N-end rule (Fig. 2a; calpain_cleavage_prediction/n-terminal_site_classes.csv; bash syntax file calpain_cleavage_prediction/syntax).

The respective proteins were classified based on overall abundance of putative DEK1 cleavage sites and prevalence of NERD signatures in the resulting N termini using a combination of principal component analysis (PCA) and model-based clustering implemented in the R packages FactoMineR and mclust[100] (Supplementary Fig. S6a–f; Jupyter R notebook calpain_cleavage_prediction/DEK1_cleavage_sites.ipynb).

Overall site frequency and individual NERD site type frequencies were scaled by total protein length. Log-transformed, scaled overall site frequencies were clustered with *k*-means clustering into five site abundance level categories (SLC; Supplementary Fig. S6a). The matrix was utilized for PCA (Supplementary Fig. S6b). The first ten principal components were used for model-based clustering using default parameters (Supplementary Fig. S6c, d). The resulting clusters were interpreted in the context of the first five principal component eigenvectors (explaining ~99% of the total variation; Supplementary Fig. S6d) and the distribution among the five SLCs (Supplementary Fig. S6e, f). We assessed the proportion of cleavages resulting in NERD-like signatures that have been experimentally confirmed *in planta* (Supplementary Fig. S6e, f).

## Tracing misregulation and predicted calpain cleavage patterns in the *P. patens* GRN

Directed network enrichment analysis (Fig. 3a) was performed with the NEAT R package[27] using the R Jupyter notebook calpain_cleavage_prediction/ NEAT_cleavage.ipynb. Ratios between observed and expected sizes of specific candidate gene sets were clustered for both rows and columns using the ward.D2 method and plotted as a heat map (Fig. 3a).

To study gene-wise effects of *DEK1* mutation, we calculated the cumulative misregulation levels of genes with significantly altered expression levels in the mutants as the sum of the three individual $\chi^2$ test statistics of the three likelihood-ratio tests (LRTs) comparing WT to $\Delta dek1$, *oex1* to WT, and $\Delta dek1$ to *oex1* employing it in the sense of an absolute, cumulative effect size.

Cumulative misregulation levels were used to analyze their patterns in the five DEK1-controlled subnetworks using complementary approaches: generalized linear modeling, random forest classification, PCA, $\chi^2$ tests and correlation analysis (Fig. 3b and Supplementary Fig. S6g, h; R Jupyter notebook calpain_cleavage_prediction/CCinRegulators.TargetPerspective. only_target_subnetworks.ipynb). Upstream regulatory context of all genes in the five DEK1-controlled subnetworks were analyzed by tracing incoming regulatory edges for each gene up to the third order (TF3 → TF2 → TF1 → gene) building on the make_e-go_graph function of the igraph R package. We repeated these analyses for the three subnetworks encoding the 2D-to-3D transition (V → II → X; Fig. 3c, d; R Jupyter notebook calpain_cleavage_prediction/ CCinRegulators.TargetPerspective.only_II_V_X.ipynb).

## Filtering the final set of direct and indirect DEK1 targets

To define direct and indirect DEK1 targets, we analyzed the initial set of 215,189 regulatory interactions from all subnetworks involving TFs as regulators (R Jupyter notebook calpain_cleavage_prediction/GetCandidateTargets.ipynb). Information on calpain cleavage classification, significance of overall misregulation in *dek1* mutants and the global *dek1* mutants misregulation pattern were added. The final set of DEK1-controlled regulatory interactions (Fig. 3e and Supplementary Fig. S7) was analyzed using the R Jupyter notebook calpain_cleavage_prediction/filter_analyze_DEK1Targets.ipynb selecting TF regulators with a classified NERD-like cleavage pattern and significantly misregulated target genes. This analysis resulted in 10,120 network edges

(Supplementary Data S6). The mosaic plot comparing the different types of DEK1-controlled regulatory interactions across the 11 subnetworks (Supplementary Fig. S7c) was created using the R Jupyter notebook calpain_cleavage_prediction/filter_analyze_DEK1Targets.ipynb. Alluvial plots were created using the R Jupyter notebook calpain_cleavage_prediction/alluvial.ipynb. Ontology term enrichment analysis (Fig. 3f, g) was carried out using the Jupyter notebook calpain_cleavage_prediction/ontologies/AnalyseEnrichment.ipynb and plotted using calpain_cleavage_prediction/ontologies/PlotSelectedTerms.ipynb. Terms shown in Fig. 3f, g were filtered for overall enriched terms among targets in subnetwork II, V, and X and for connection to *dek1* phenotypes. Text size was scaled by the number of genes per GO term and gene set. Results were combined for overall enrichment among DEK1 targets and subnetwork interconnection sets.

## Functional characterization of DEK1 targets

Primary ontology term enrichment analysis for the DEK1 targets was carried out using the ontology_enrichment_workflow as described above for the DGE sets. We carried out two types of comparisons looking globally at all targets as well as at individual subnetwork pairings and deregulation patterns (e.g., activator_II_vs_X or repressor_X_V). The enriched terms at FDR < 0.1 for each partition of GO and PO was analyzed and plotted using the R Jupyter notebook calpain_cleavage_prediction/ontologies/AnalyseEnrichment.ipynb.

As already demonstrated for the overall DGE sets, DEK1 mutation results in a deregulation of broad gene functions. Thus, our goal was to identify overall trends without losing specificity. We chose a two-pronged approach, combining automated semantic analysis with manual identification of representative key concepts.

The basis for the automated analysis of enriched ontology terms, is semantic similarity analysis[101] and information content- or ontology-based ranking of similar concepts. We used the ontologyX suite of R packages[102] in the above-mentioned R Jupyter notebook to compare and cluster entire gene sets (Supplementary Fig. S8a–c) as well as compare, group and rank individual enriched terms to select the most informative for plotting their semantic similarity-derived distance matrix (1-S) via multidimensional scaling (Supplementary Fig. S8d, e).

The result of this automated analysis was then manually inspected, interpreted and curated in light of the external knowledge of DEK1 mutant phenotypes in *P. patens* and other plants, calpains as well as the NERD pathway to select the most informative and non-redundant concepts (Fig. 3f, g; R Jupyter notebook calpain_cleavage_prediction/ontologies/ PlotSelected Terms.ipynb).

## Factorial Differential Gene Expression Network Enrichment Analysis (FDGENEA)

Phenotypic observations of the *dek1* mutant lines (Supplementary Figs. S9 and S10a) were translated into 17 binary, factorial variables (.tsv file fdgenea/ phenotypic_factors.tsv; Fig. 4a and Supplementary Fig. S10b) and used for DGE analysis using the R sleuth package (R Jupyter notebook fdgenea/ Analysis.*.ipynb) as described above. A positive association between a gene and a trait corresponded to a Wald's test *b*-value > 0, with negative associations for *b* < 0. The LRT test statistic was interpreted as an effect size for the strength of the association. The full data are provided in the files fdgenea/ dge/*/dge.full.tsv.gz.

Subsequently, for each trait, network enrichment analysis of significantly associated genes (LRT FDR < 0.01) was conducted using the NEAT R package (R Jupyter notebook fdgenea/NEAT.iypnb). *p* values were adjusted for multiple testing using the Benjamini-Hochberg method and filtered at 99% confidence. Two types of NEAT analyses were carried out:

(1) Testing enrichment of the two directional sets of significantly associated genes independently (e.g., trait number_buds_per_filament comparison of normal vs. high is up, i.e., genes with *b* > 0). Full results of this analysis are provided in the .tsv file fdgenea/ NEAT_subnetwork_enrichment.phenotypic_factors.tsv;

(2) Testing enrichment of the entire set of genes that is significantly associated with a given trait (e.g., trait number_buds_per_filament comparison of normal vs. high, i.e., genes with $b < 0$ or $b > 0$). Full results of this analysis are provided in the .tsv file fdgenea/NEAT_subnetwork_enrichment.phenotypic_factors.simplified.tsv.

The results of these two analyses were combined for Supplementary Fig. S10a, using the second analysis as a basis for the heatmap and the results of the first to derive the cell annotations ($+/-$), to indicate significantly enriched sets of either positively (up, i.e., $+$) or/and negatively (down, i.e. $-$) associated genes for any of the subnetworks.

As a final step, we implemented a procedure that analyzed the network structure of the genes associated with each trait, isolated subgraphs with enriched trait association, and identified common upstream TF genes. For each gene, the algorithm evaluates the cumulative effect size (i.e., the association of the downstream genes with the trait) and records the numbers of all downstream regulators and only the downstream TFs in both cases distinguishing between direct (first-order) downstream genes and indirect (higher-order) downstream genes. The cumulative effect size was recorded both as a sum of the LRT test statistic (columns ending in _cb) and as the sum of the absolute value of the LRT test statistic (columns ending in _cab). The identified connected components were reported as .tsv files with all the collected statistics as well as individual GraphML files that can be opened in Cytoscape. The algorithm was applied to two datasets: one comprising all subnetworks and the other comprising only the NEAT-enriched (FDR < 0.01) subnetworks for each trait. The algorithm was implemented in R using the igraph package (Multi-threaded R code fdgenea/FDGENEAE_all.R and fdgenea/FDGENEA_only_enriched.R). An example of the procedure with additional plots and analyses are provided for the overbudding trait (normal vs. high number_buds_per_filament; Fig. 4) in the R Jupyter notebook fdgenea/ FDGENEA.overbudding_only.ipynb.

The GraphML of the identified connected components for both datasets for the overbudding phenotype (Fig. 4a) as well as subgraphs/regulons of identified key players were analyzed and plotted using Cytoscape (Figs. 4a, c and S10c, d, i–m).

Intersections among the FDGENEA sets for all traits were analyzed in the R Jupyter notebook fdgenea/Intersections.ipynb (Fig. S10e, f).

Intersections among the FDGENEA sets for all traits and the two types of predicted DEK1 targets (activator targets; *repressor targets*) were analyzed in the R Jupyter notebook fdgenea/DEK1_targets.Intersections.ipynb (Supplementary Fig. S10g, h).

### Further characterization of overbudding genes using cell-type specific transcriptomes

Significant DEGs from the cell-type specific transcriptome data for protonemal tip cells and bud cells[23] were mapped to the current genome annotation, intersected with the predicted direct and indirect DEK1 targets as well as the genes associated with the overbudding phenotype and plotted as an alluvial plot (Fig. 4b) using the R Jupyter notebook fdgenea/Cell-type_transcriptomes.Intersections.ipynb. The partial overlap between DEK1-controlled AP2 and MYB TFs in subnetworks II (Supplementary Fig. S10l, m) was analyzed using Cytoscape and intersections were drawn using the VIB Venn web interface (http://bioinformatics.psb.ugent.be/webtools/Venn).

### Reporting summary

Further information on research design is available in the Nature Portfolio Reporting Summary linked to this article.

### Data availability

The *Physcomitrium patens* line (*oex1*) generated in this study as well as other *P. patens* lines used have been deposited at Comenius University in Bratislava, Department of Plant Physiology moss collection, and are listed in Supplementary Data sheet S13. RNA-seq data have been deposited at EBI Array Express and are publicly available as of the date of publication (E-MTAB-10907). All generated data sets have been deposited at Zenodo https://doi.org/10.5281/zenodo.5513495. This paper analyzes existing, publicly available data. A table with all accession numbers for public datasets is provided in Supplementary Data sheet S13. Raw images generated in this study, including microscopy, gel and immunoblot images, are publicly available as part of the Zenodo archive and listed in Supplementary Data sheet S13. All 27 *P. patens* gene sets used in the figures or the text are provided as gene id lists in plain text files in the gene_sets/ folder of the Zenodo archive listed in Supplementary Data sheet S13. Postgresql table dumps, as well as additional .tsv/.csv tables that are not explicitly mentioned in the text below but are used in the Jupyter notebooks, are provided in the Zenodo archive listed in Supplementary Data sheet S13. If not listed explicitly in the "Methods" section or Supplementary Data sheet S13, data files underlying each Figure are defined in the respective Jupyter notebook and are uploaded as part of the Zenodo archive and github repository (see below). The corresponding Jupyter notebook for each Figure is described in the "Methods" section.

### Code availability

All original code has been deposited to git repositories. Parallelized Snakemake workflows are provided as individual repositories. Data analyses, statistics and visualizations were implemented via R or Python Jupyter Notebooks and for convenience are also accessible via a GitHub repository (https://github.com/dandaman/moss_DEK1_GRN_analysis). All git repositories have been pushed to GitHub and deposited at Zenodo and are publicly available. DOIs are listed in Supplementary Data sheet S13. Used packaged software are provided via conda environments included in the Zenodo archive listed in Supplementary Data sheet S13. File names of the environments correspond to the Jupyter kernels of each notebook.

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

## Acknowledgements

The research reported herein was made possible by FRIMEDBIO grant 240343 from the Norwegian Research Council to O.A.O. Sequencing services were provided by the Norwegian Sequencing Centre (www.sequencing.uio.no), a national technology platform hosted by the University of Oslo and supported by the Functional Genomics and Infrastructure programs of the Research Council of Norway and the Southeastern Regional Health Authorities. V.D. was supported by FRIMEDBIO grant from the Norwegian Research Council and the Slovak Research and Development Agency grants APVV-17-0570 and APVV-21-0227. The authors greatly appreciate the input from two anonymous reviewers whose comments and suggestions were helpful to finalize the manuscript.

## Author contributions

O.A.O. initiated and designed the *P. patens* mutant study and contributed to writing the manuscript. V.D. carried out the *P. patens* wet lab work including microscopy, observations and interpretations, contributed to the phenotypic factor annotations, performed RNA extraction, genotyped the *oex1* line, prepared Fig. 1b, c and Supplementary Figs. S9 and S10a and contributed to manuscript writing. T.B. performed initial computational analysis of the *DEK1* RNA-seq data. M.M. performed QC of the *DEK1* RNA-seq data and contributed to the initial *DEK1* RNA-seq analysis. T.H. contributed to the initial computational analysis of the *DEK1* RNA-seq data. P.F.P. contributed to the initial *DEK1* RNA-seq data analysis, created the *oex1* line and contributed to the phenotypic factor annotations. K.F.X.M. contributed to writing the manuscript. W.J. and A.E.A. contributed to molecular characterization of the *oex1* line and preparation of RNA samples used for the RNA-seq analysis. D.L. conceived, designed, implemented and carried out all presented data analyses (final *DEK1* RNA and global RNA-seq analyses, GRN inference and analysis, calpain target predictions, phylogenomics, ontology and other functional annotations, enrichment analyses, FDGENEA), postulated the DEK1-NERD hypothesis, wrote software, created and curated data sets, prepared the figures and wrote the manuscript.

## Funding

## Competing interests

The authors declare no competing interests.
