## [Peer review file · Communications Biology]

Reviewers' comments:

Reviewer #1 (Remarks to the Author):

As a calpain cysteine protease, DEK1 has been quite well characterized and is implicated in development and cell fate transitions both in animals and plant systems. In the moss, the *dek1* knock-out mutant exhibited an impaired 2D-to-3D transition and failed to form any mature gametophore. Instead, the mutant was characterized by overly proliferated buds. This has led to the suggestion that DEK1 may be involved in cell fate transition but solid evidence for this is missing as substrates downstream to DEK1 are not known. The manuscript from Demko et al attempted to address this piece of missing puzzle by performing bulk RNA-seq on WT, a null allele (*dek1* knock-out), a hypermorph allele (*oex1*) and 2 hypomorph alleles (Δ loop, Δ lg3) at different time points. The authors then performed a tour de force of bioinformatics analysis to establish the multi-tier gene regulatory network which could be regulated by DEK1. As an experimentalist, although I lack the expertise to comment on the technical details, I do think bioinformatics outputs of this paper are of outstanding qualities and should be of immense interest for developmental biologists who are interested in the 2D-to-3D transition process. There are however some inadequacies which I think the authors need to address before the manuscript can be accepted for publication:

Major

(1) The authors have used the Kallisto/Slueth package for their DEG analyses. Can the authors justify their choice of workflow? As far as I am aware there are far more popular/mainstream packages such as DEGseq and EBSeq for DEG identification. In another word, can the authors use an independent approach to confirm that DEGs identified are valid. This is particularly important as the downstream GRN was established based on these DEGs. I would expect a good portion (~70-80%) of the DEGs identified using Kallisto/Slueth be reproduced in the independent method for them to be considered valid.

(2) Lack of direct evidence to approve/disapprove the predictive power of their analyses. I appreciate the power of big data analyses and the innovative FDGENEA approach devised by the authors to identify potential DEK1 targets that are responsible for the phenotype observed. However, the question still remains as to how reliable are these predictions and can the authors do more to substantiate their claims? For example, can the author check the protein abundance of predicted DEK1 direct target (with Calpain cleavage sites) in the Δ dek1 background to confirm its degradation is indeed impaired? If an antibody for such a direct target is not available, tagged (e.g. GFP fusion) knock-in constructs can be introduced into the Δ dek1 to facilitate immunodetection.

(3) Their subnetwork analyses show that DEK1 lies downstream of APB2 and APB1 in subnetwork II. However, in another RNAseq experiment that compared the transcriptomes between WT and Δ dek1, APB2 and APB3 were found to be upregulated in Δ dek1 (Demko et al., 2014), suggesting that APB2 may itself be subjected to the transcriptional regulation of a DEK1-targeted activator, and hence is downstream to DEK1. I wonder do the authors have any explanations to resolve this discrepancy? At the same time, can the authors show that DEK1 expression is altered in the *apb* knock-out alleles if their prediction is correct?

(4) In Figure 3b, c, the authors claim that target gene misregulation is positively correlated with upstream TFs. Please include correlation coefficient r^2 values.

(5) The authors have dedicated Figure 4 to be part of the discussion. It would have been more appropriate if this and the relevant writing are moved to results. It is especially important given the last part of Results does read like an abrupt stop.

Legends for figure3 and figure 4 are unusually long! They could have been the longest I have ever read. I think the authors should use the legend wisely to help the reader understand the figure better. For me, large part of the legend is actually method and data interpretation. There are also missing references in Figure 3.

(6) Although the figures are visually pleasing, the poor layout of this manuscript has made reviewer's job extremely difficult. There were no page number and line number-how am I supposed to make reference to certain parts of the manuscript? Unexplained underlined text throughout the figure legend-what do they mean??

Minor

(1) Fig.1b- arrow misplaced in Δ dek1 and oex1. Remove shadow for arrows and error bars as they are very distracting

(2) Fig3 legends

Fig3f, g – I find it difficult to appreciate the word clouds format. It may appear fashionable but I think the good old bubble plot or bar chart with easily extractable parameters (p-values and number of genes) are easier to read and more organized from a reader perspective.

(3) DEK1 and dek1 were used interchangeably. For example:

-.....in these conserved developmental processes and their role in the pleiotropic DEK1 phenotype..... \diamond should be dek1

-DEK1 dramatically affects moss development \diamond should be dek1

Typo

(1) DEK1 dramatically affects moss development-should be "loss of DEK1..."

(2) Ancestral function of the calpain superfamily is cell division and cell cycle regulation- should be "Ancestral functions of the calpain superfamily are"

(3) Allele nomenclature-

At first glance, I thought Δ loop indicate a DEK1 overexpressor with a loop deletion. I strongly suggest the author to follow the original nomenclature of different alleles to avoid confusion.

Δ loop \diamond should be dek1 Δ loop

Δ Ig3 \diamond should be dek1 Δ Ig3

(5) "We performed differential gene expression (DGE)..." -should be DEG

(6) For instance, these genes were involved in the biological

(7) Time cours- should be time course

References

(1) Importantly, the NERD pathway components were recently identified in *P.patens* and mutants in a key component found to arrest 2D-to-3D transition⁴⁷-sholuld this be Moody et al 2018? At the moment it is Kucera et al., a paper on Cytoscape app.

(2) Refence numbering, no. 4 was repeated twice

Reviewer #2 (Remarks to the Author):

In this manuscript, the authors extensively analyzed the gene regulatory network structure to investigate the roles of calpain DEK1 in cell fate transitions during the 2D-to-3D development of the moss *Physcomitrium patens*. Using a combination of meticulous analysis of phenotypic traits, comprehensive transcriptomics, and data science methods, they developed a model that highlights the role of DEK1 as a post-translational regulator of gene expression and proposes its pivotal role in regulating cell fate transitions. This study provides a novel approach to investigate plant development and significantly contributes to the field of plant science by advancing our understanding of the intricate mechanisms underlying cell fate transitions. However, there are several concerns that need to be addressed.

1) Overall, the order of the results presented in the text does not align with the order of the corresponding figures, which makes it difficult to follow the reasoning behind the results. It would be better to rearrange the figures to match the flow of the text, improving the coherence of the manuscript.

2) In Fig.1, considering the authors' aim to identify differentially expressed genes between the wild type and dek1 mutants for subsequent gene regulatory network analyses, it should be crucial to minimize the inclusion of false positive genes during the initial gene extraction stage. Therefore, the authors should consider employing a more stringent criterion, such as FDR < 0.01 for the RNA-

seq data analysis, instead of $FDR < 0.1$.

3) In Fig. S6, the figures labeled as d, e, and f are missing, which makes it difficult to evaluate the predicted target transcription factors expected to be cleaved by DEK1. It is necessary to include these missing figures in order to provide a complete understanding of the predicted cleavage targets and their implications.

4) The absence of proper citations for Figures or Supplementary data in the text hinders the understanding of the context. For instance, on page 6, where it is mentioned that "we compiled 374 public and novel RNA-seq libraries and 1,736 novel annotated regulators using the random forest predictor of GENIE3", it is crucial to specify the source or location of these public RNA-seq libraries in the text, allowing readers to access the relevant data. Additionally, on page 7, "The upstream regulatory context suggested that DEK1 expression is induced by subnetwork V TF genes and activated early in development.", Is the data supporting this statement from Fig. S5a? Conversely, Fig. 2b-f, Fig. 4a-c, S2c, S2d, and S5a are not cited in the Results and Discussion section. It is crucial to provide proper citations indicating which figure each result is based on, as well as to provide clear descriptions of the contents and significance of each figure.

5) It would be helpful, if possible, to include the complete set of subnetworks labeled from I to XI in supplementary figures. Additionally, providing information about the typical transcription factors found in each subnetwork would enhance understanding and interpretation of the data.

6) Fig. 1 includes the phenotypes of Δ loop and Δ lg3; however, since these mutants were not utilized in the subsequent transcriptome and GRN analyses, it is better to remove these data from the text to maintain clarity and focus on the relevant analyses.

Minor comments:

7) In Fig. 1c, the authors need to specify the number of observed filaments.

8) To facilitate commenting and referencing, the authors should add page and line numbers in the text.

9) In Fig. S5, there are two S5a. One of the two should be corrected to S5b.

10) The citation #47 mentioned on page 7 in relation to the NERD pathway components appears to be incorrect. It should be verified and corrected accordingly.

11) In Fig. S6a legend, the definitions of S1 through S4 (no, very few - few, few -medium, medium - many, and many - very many) are vague and require clarification to provide a better understanding of these terms.

12) The text contains remnants of corrections, so it is necessary to make corrections.

Reviewer #1 (Remarks to the Author):

As a calpain cysteine protease, DEK1 has been quite well characterized and is implicated in development and cell fate transitions both in animals and plant systems. In the moss, the *dek1* knock-out mutant exhibited an impaired 2D-to-3D transition and failed to form any mature gametophore. Instead, the mutant was characterized by overly proliferated buds. This has led to the suggestion that DEK1 may be involved in cell fate transition but solid evidence for this is missing as substrates downstream to DEK1 are not known. The manuscript from Demko et al attempted to address this piece of missing puzzle by performing bulk RNA-seq on WT, a null allele (*dek1* knock-out), a hypermorph allele (*oex1*) and 2 hypomorph alleles (\square loop, \square lg3) at different time points. The authors then performed a tour de force of bioinformatics analysis to establish the multi-tier gene regulatory network which could be regulated by DEK1. As an experimentalist, although I lack the expertise to comment on the technical details, I do think bioinformatics outputs of this paper are of outstanding qualities and should be of immense interest for developmental biologists who are interested in the 2D-to-3D transition process. There are however some inadequacies which I think the authors need to address before the manuscript can be accepted for publication:

RE: Dear reviewer #1 thank you very much for taking the time to carefully evaluate our manuscript and for helpful comments and suggestions to improve our manuscript. Please find our responses below (text prepended with RE#.#:).

Major

(1) The authors have used the Kallisto/Sleuth package for their DEG analyses. Can the authors justify their choice of workflow? As far as I am aware there are far more popular/mainstream packages such as DEGseq and EBSeq for DEG identification. In another word, can the authors use an independent approach to confirm that DEGs identified are valid. This is particularly important as the downstream GRN was established based on these DEGs. I would expect a good portion (~70-80%) of the DEGs identified using Kallisto/Sleuth be reproduced in the independent method for them to be considered valid.

RE1.1: In fact we have assessed and tested multiple DGE tools in the analysis of these dataset before finally settling on the chosen Kallisto/Sleuth workflow. We have left this out of the manuscript to not further add to the “tour the force of bioinformatics” i.e. readability and length of the manuscript and supplement. Besides the chosen workflow, we also assessed the more common, mapping-based approaches (read mapping with HISAT2 then HTseq or feature-count coupled with DGE analysis in edgeR or DGESeq2) requiring mapping of reads to the genome prior to read counting. In the end, in this setting we opted for the alignment-free workflow of kallisto based on the following rationale:

- a) Successful applications and best performance in qPCR- and other validations in other plant gene-expression analyses, we were involved in, e.g. the bread wheat transcriptome (10.1126/science.aar6089).
- b) In a redundant plant genome with many repetitive and pseudoallelic gene loci (10.1126/science.1150646, 10.1186/1471-2164-14-498) and frequent non-canonical splicing (10.1186/1471-2229-10-76, 10.1093/nar/gky225) we often encountered specific problems (see the mentioned literature for examples) with pipelines comprised of spliced-alignment (e.g. HISAT2 or bwa), read count (e.g. cufflinks, feature-count or HTSeq) and DGE tools (DGESeq2 or edgeR) developed for mammals with a substantially divergent gene and genome structure. In such a setup with multiple tools many parameters affect the outcome and need to be fine-tuned to achieve the optimal tradeoff between sensitivity and specificity. We wanted to employ direct gene-level RNASeq quantification on a non-redundant transcriptome which we specifically established for this study (see supplementary materials). The Kallisto/Sleuth framework offers a direct way to quantify gene-wise abundance by modeling kmer-based pseudo-counts on potentially multiple transcript

isoforms and condensing them into a genic expression level estimate that were used both for bootstrapped DGE analysis with sleuth and VST-transformed to be used in downstream coexpression analyses and regulatory network prediction.

- c) In our comparison with the aforementioned mapping-based tools based on the existing qPCR data for \square dek1 (10.1104/pp.114.243758) kallisto/sleuth came in on top and (at least at that time) was the only tool that offered a direct way to model time-series data in combination with additional factors using GLM. This enabled us to assess genes that differ in their levels along the time course in *one* comparison and thus substantially reduces the number of comparisons to present/discuss/understand.
- d) As the only tool sleuth offers a direct way to utilize the technical bootstrap replicates generated by kallisto resulting in an additional quality measure/filter and offers two alternative approaches for significance testing (likelihood ratio test and Wald test) which we both applied in intersection to define significantly differentially expressed genes (see below for cutoffs and Methods/Supplement for methodological details).

RE1.2: Reading your remarks regarding the DGE analyses made it clear to us that our description of the carried out analyses and cutoffs in the main text were insufficient. Allow me to clarify:

- a) The network analyses were carried out on the full gene set and only subsequently intersected with the different DGE result sets. Thus, these results do not depend on the chosen cutoff of the DGE analysis
- b) We have carried out both a strict ($qval < 0.01$ /FDR 1%) and a relaxed ($qval < 0.1$ /FDR 10%) q value cutoff and provide both results in numbers and gene sets as part of the supplement Supplementary Table S1 (XLSX) and Key Resource Table in the Supplementary Material (PDF) for name/description of the set files as part of the ZENODO archive (<https://doi.org/10.5281/zenodo.5513496>). DGE analysis is only one step in a multi-step data mining workflow we have applied to define the final sets of candidate indirect DEK1 targets. We have long debated which of these numbers to use for the main text and figures, because in such a multiple-step data mining approach, applying too strict cutoffs in the individual analysis steps inadvertently results in substantial loss of true-positives. We greatly appreciate and agree with your assessment that the resulting gene sets are of immense interest for developmental biologists who are interested in the 2D-to-3D transition process. From this experimentalist perspective, in our opinion, a more comprehensive gene list optimizing the ratio between false-positives/false-negatives due to intersection of multiple analysis steps and opting for the discussion of only one cutoff per analysis step in the main article, is preferable. Furthermore, each supplemental table always contains the full set of genes (reversibly restricted to the cutoffs discussed in the text using the xlsx auto-filter). Thus, readers always have the option to look up p -values, q -values etc. from multiple methods for their gene of interest and evaluate them individually. Thus, inspired by your comment, to avoid confusion and still maximize this utility, we chose to alter the text describing the results of the DGE analysis and include a brief discussion of the strict vs relaxed cutoff and introduce better references to the full results in the supplement.

RE1.3: As mentioned above, we have tested several approaches. As an example for the consistency with those, we include a comparison of kallisto/sleuth with DGE analyses carried out using edgeR (new supplementary Fig S12). As only sleuth supported time series modeling, we compared two strategies for edgeR: 1) pooling all time points per genotype and 2) testing time points individually and merging them prior to set analyses. The consistency between both approaches is very well in the range you expected (e.g. ~76% of sleuth DEGs in comparing up-regulated genes in the DEK1 null mutant).

(2) Lack of direct evidence to approve/disapprove the predictive power of their analyses. I appreciate the power of big data analyses and the innovative FDGENEA approach devised by the authors to identify potential DEK1 targets that are responsible for the phenotype observed.

However, the question still remains as to how reliable are these predictions and can the authors do more to substantiate their claims? For example, can the author check the protein abundance of predicted DEK1 direct target (with Calpain cleavage sites) in the \square dek1 background to confirm its degradation is indeed impaired? If an antibody for such a direct target is not available, tagged (e.g. GFP fusion) knock-in constructs can be introduced into the \square dek1 to facilitate immuno-detection.

RE2: While we agree that experimental confirmation of our hypothesis is the next step, the suggested work would go beyond the scope of this comprehensive computational analysis. Nevertheless, our predictions of regulatory subnetworks as well as TF-target, genotype-phenotype and (indirect) DEK1-target interactions (Supplementary Tables S4, S6, S8, and S10) are already well supported by existing experimental work in *Physcomitrella*, *Arabidopsis* and other plants. To illustrate this, we have added a series of citations for individual examples to the introduction, results and discussion (adding four new paragraphs) as well as a description of the high consistency with tip-and bud-cell specific gene lists identified previously (Figures 4b, S10 and Supplementary Table S11).

(3) Their subnetwork analyses show that DEK1 lies downstream of APB2 and APB1 in subnetwork II. However, in another RNAseq experiment that compared the transcriptomes between WT and \square dek1, APB2 and APB3 were found to be upregulated in \square dek1 (Demko et al., 2014), suggesting that APB2 may itself be subjected to the transcriptional regulation of a DEK1-targeted activator, and hence is downstream to DEK1. I wonder do the authors have any explanations to resolve this discrepancy? At the same time, can the authors show that DEK1 expression is altered in the apb knock-out alleles if their prediction is correct?

RE3: Thank you for pointing out this observation! In fact, it is an argument for our hypothesis that we obviously did not explain well enough. We have added a paragraph presenting S5a and S5b that demonstrate the APB regulatory hierarchy and DEK1 upstream regulon. Not only provide these two figures several examples of already confirmed regulatory links aside from the APBs (see new text), but they also help to address your question: APB-2, APB-3 and APB-1 are downstream of APB-4 which we predict to be a direct DEK1 target. Hence if the control of APB-4 by DEK1 is missing in \square dek1 we would expect ectopic expression of these APBs (1-3) leading to the observed over-budding phenotype and others discussed in the text. While APB-1 and APB-2 are in the order-filtered upstream regulatory context of DEK1, APB-3 would be as well, if we increased the order (number of intermediary links). From the perspective of DEK1, APB-3 is higher up the hierarchy than APB-2/1. Possibly the three genes address different aspects/roles in the 2D/3D transition. APB-3 e.g. is part of the subgraph of deeply-conserved DEK1-controlled, overbudding up-regulated genes and their top5 DEK1-controlled regulators (S10K) that is in control of the CLV-controlled cell-wall/polarity aspects (likely in the asymmetrical divisions during bud formation) also described in the *Arabidopsis* stem cells with DEK1 phenotypes. APB-2/1 seem to affect more the auxin/ent-kaurene signaling aspects that are clearly linked to the formation of the sub-apical cell identity in *Physcomitrium* (possibly determined by phytohormone gradients involved). Thus, they may play a role in forming different states/cellular identities (Fig 4f). In our model, DEK1 is the fine-tunable encoded off-switch of these regulons. If the levels of these TFs increase, so does DEK1 and thus increasing the chances for auto-catalytic release of the specific or unconstrained calpain (Fig 4d) switching them off again.

text throughout the (4) In Figure 3b, c, the authors claim that target gene misregulation is positively correlated with upstream TFs. Please include correlation coefficient r^2 values.

RE4: We have inserted the global Pearson correlation coefficient ρ in the main text. The full statistical analysis including more plots, correlation tests, generalized linear regression analysis and ML-based classification of significant deregulation is presented in the supplementary data e.g. CCinRegulators.TargetPerspective.only_target_subnetworks.ipynb available from the ZENODO archive or browsable in the github repo:

https://github.com/dandaman/moss_DEK1_GRN_analysis/blob/main/calpain_cleavage_prediction/CCinRegulators.TargetPerspective.only_target_subnetworks.ipynb

(5) The authors have dedicated Figure 4 to be part of the discussion. It would have been more appropriate if this and the relevant writing are moved to results. It is especially important given the last part of Results does read like an abrupt stop.

RE5.1: You're absolutely correct. We have substantially extended the text adding two new paragraphs to present Fig 4 and also link to existing experimental data from the moss and other plants that supports our predictions.

Legends for figure3 and figure 4 are unusually long! They could have been the longest I have ever read. I think the authors should use the legend wisely to help the reader understand the figure better. For me, large part of the legend is actually method and data interpretation. There are also missing references in Figure 3.

RE5.2: Yes the legends are long, but these are complex composite figures with a lot of information encoded by formatting, shape and coloring etc. that need to be described for the reader to follow our conclusions drawn from them. However, where appropriate, we have moved parts to the methods and to the discussion.

(6) Although the figures are visually pleasing, the poor layout of this manuscript has made reviewer's job extremely difficult. There were no page number and line number-how am I supposed to make reference to certain parts of the manuscript? Unexplained underlined figure legend-what do they mean??

RE6: We apologize for this oversight in the previous ms. version! We have added page and line numbering. We have removed the underline formatting in the legend texts.

Minor

(1) Fig.1b- arrow misplaced in □dek1 and oex1. Remove shadow for arrows and error bars as they are very distracting

RE.M1: The misplaced arrows were repositioned and shadows were removed.

(2) Fig3 legends

Fig3f, g – I find it difficult to appreciate the word clouds format. It may appear fashionable but I think the good old bubble plot or bar chart with easily extractable parameters (p-values and number of genes) are easier to read and more organized from a reader perspective.

RE.M2: The plot you suggest would be either unreadable or would require a full page. However, the data you request are provided in the supplementary table S7.

(3) DEK1 and dek1 were used interchangeably. For example:

-.....in these conserved developmental processes and their role in the pleiotropic DEK1 phenotype..... □ should be dek1

-DEK1 dramatically affects moss development □ should be dek1

RE.M3: A use of DEK1 for a protein, *DEK1* for a gene, and *dek1* for a mutant has been unified throughout the text.

Typo

(1) DEK1 dramatically affects moss development-should be “loss of DEK1...”

RE.T1: The title has been modified to “Loss of DEK1...” as suggested.

(2) Ancestral function of the calpain superfamily is cell division and cell cycle regulation- should be “Ancestral functions of the calpain superfamily are”

RE.T2: corrected

(3) Allele nomenclature-

At first glance, I thought Δ loop indicate a DEK1 overexpressor with a loop deletion. I strongly suggest the author to follow the original nomenclature of different alleles to avoid confusion.

Δ loop Δ should be dek1 Δ loop

Δ lg3 Δ should be dek1 Δ lg3

RE T3: Allele nomenclature for *dek1* mutants has been modified as suggested.

(5) “We performed differential gene expression (DGE)...” -should be DEG

RE.T5: Our usage of these acronyms is intentional. The common usage of this terminology e.g. used in all method descriptions of edgeR, DESeq2, kallisto etc is: method: Differential Gene Expression analysis (DGE) <-> outcome: Differentially Expressed Genes (DEG). We have used and explained both acronyms accordingly.

(6) For instance, these genes were involved in the biological

Re.T6: corrected

(7) Time cours- should be time course

Re.T7: corrected

References

(1) Importantly, the NERD pathway components were recently identified in *P.patens* and mutants in a key component found to arrest 2D-to-3D transition⁴⁷-should this be Moody et al 2018? At the moment it is Kucera et al., a paper on Cytoscape app.

(2) Refence numbering, no. 4 was repeated twice

RE.R: Reference numbering was corrected in both instances

Reviewer #2 (Remarks to the Author):

In this manuscript, the authors extensively analyzed the gene regulatory network structure to investigate the roles of calpain DEK1 in cell fate transitions during the 2D-to-3D development of the moss *Physcomitrium patens*. Using a combination of meticulous analysis of phenotypic traits, comprehensive transcriptomics, and data science methods, they developed a model that highlights the role of DEK1 as a post-translational regulator of gene expression and proposes its pivotal role in regulating cell fate transitions. This study provides a novel approach to investigate plant development and significantly contributes to the field of plant science by advancing our understanding of the intricate mechanisms underlying cell fate transitions. However, there are several concerns that need to be addressed.

RE: Dear reviewer #2 thank you very much for taking the time to carefully evaluate our manuscript and for helpful comments and suggestions to improve our manuscript. Please find our responses below (response text prepended with RE#.#:).

1) Overall, the order of the results presented in the text does not align with the order of the corresponding figures, which makes it difficult to follow the reasoning behind the results. It would be better to rearrange the figures to match the flow of the text, improving the coherence of the manuscript.

RE1: We have gone over the entire text to ensure that the figures are cited in order and adjusted the text where appropriate. The most tricky situation (and possibly the first instance you've noticed) is Fig. 1e which is cited before 1b-1d. This has also been a concern when we designed the composite figure. Due to the different sizes of the composites, the placement of 1e is restricted. More importantly it also works as a legend for Fig. 1f as it introduces the activator/repressor color-coding. Thus, we decided to align the figure as presented as it allows for most of the panels to be in order of citation and context. Another situation you probably meant was Figure 4. This has been addressed while adding additional result paragraphs.

2) In Fig.1, considering the authors' aim to identify differentially expressed genes between the wild type and *dek1* mutants for subsequent gene regulatory network analyses, it should be crucial to minimize the inclusion of false positive genes during the initial gene extraction stage. Therefore, the authors should consider employing a more stringent criterion, such as $FDR < 0.01$ for the RNA-seq data analysis, instead of $FDR < 0.1$.

RE2: Reading your remarks regarding the DGE analyses made it clear to us that our description of the carried-out analyses and cutoffs in the main text were insufficient. Allow me to clarify: The network analyses were carried out on the full gene set and only subsequently intersected with the different DGE result sets. Thus, these results do not depend on the chosen cutoff of the DGE analysis

We have carried out both a strict ($qval < 0.01$ /FDR 1%) and a relaxed ($qval < 0.1$ /FDR 10%) q value cutoff and provide both results in numbers and gene sets as part of the supplement Supplementary Table S1 (XLSX) and Key Resource Table in the Supplementary Material (PDF) for name/description of the set files as part of the ZENODO archive (<https://doi.org/10.5281/zenodo.5513496>). DGE analysis is only one step in a multi-step data mining workflow we have applied to define the final sets of candidate indirect DEK1 targets. We have long debated which of these numbers to use for the main text and figures, because in such a multiple-step data mining approach, applying too strict cutoffs in the individual analysis steps inadvertently results in substantial loss of true-positives. We greatly appreciate and agree with your assessment that the resulting gene sets are of immense interest for developmental biologists who are interested in the 2D-to-3D transition process. From this experimentalist perspective, in our opinion, a more comprehensive gene list optimizing the ratio between false-positives/false-negatives due to intersection of multiple analysis steps and opting for the discussion of only one cutoff per analysis step in the main article, is preferable. Furthermore, each supplemental table always contains the full

set of genes (reversibly restricted to the cutoffs discussed in the text using the xlsx auto-filter). Thus, readers always have the option to look up p-values, q-values etc. from multiple methods for their gene of interest and evaluate them individually. Thus, inspired by your comment, to avoid confusion and still maximize this utility, we chose to alter the text describing the results of the DGE analysis and include a brief discussion of the strict vs relaxed cutoff and introduce better references to the full results in the supplement.

3) In Fig. S6, the figures labeled as d, e, and f are missing, which makes it difficult to evaluate the predicted target transcription factors expected to be cleaved by DEK1. It is necessary to include these missing figures in order to provide a complete understanding of the predicted cleavage targets and their implications.

4) The absence of proper citations for Figures or Supplementary data in the text hinders the understanding of the context. For instance, on page 6, where it is mentioned that "we compiled 374 public and novel RNA-seq libraries and 1,736 novel annotated regulators using the random forest predictor of GENIE3", it is crucial to specify the source or location of these public RNA-seq libraries in the text, allowing readers to access the relevant data. Additionally, on page 7, "The upstream regulatory context suggested that DEK1 expression is induced by subnetwork V TF genes and activated early in development.", Is the data supporting this statement from Fig. S5a? Conversely, Fig. 2b-f, Fig. 4a-c, S2c, S2d, and S5a are not cited in the Results and Discussion section. It is crucial to provide proper citations indicating which figure each result is based on, as well as to provide clear descriptions of the contents and significance of each figure.

RE4.1: You are correct. We went over the text to improve references to the supplementary materials. As an example, we now point the reader to the Key Resources Table S2 in the sentence about the RNASeq libs and regulators.

RE4.2: We have extended the specific paragraph and also specifically reference Figures S5a and S5b.

RE4.3: We have introduced appropriate references and for some added whole paragraphs describing the results presented in the mentioned main and supplemental Figures.

5) It would be helpful, if possible, to include the complete set of subnetworks labeled from I to XI in supplementary figures. Additionally, providing information about the typical transcription factors found in each subnetwork would enhance understanding and interpretation of the data.

RE5: The full description of this analysis e.g. in context of the moss' GRN is equally if not more complex than the currently presented work and clearly goes beyond the scope and text requirements of this manuscript. We intend to publish this as a separate manuscript which we do not want to harm. Furthermore, as you correctly pointed out we need to "crucial to provide proper citations ..., as well as to provide clear descriptions of the contents and significance of each figure". Adding this merely as supplementary without discussion in the text is not advisable. Thus, we decided to limit our analysis here to the DEK1-controlled subnetworks but provide the information you mention in the supplementary tables (S4, S6, S10) and *all* data including the full network and subnetworks, annotations etc. including the figures as part of the ZENODO archive and github repository.

6) Fig. 1 includes the phenotypes of Δ loop and Δ lg3; however, since these mutants were not utilized in the subsequent transcriptome and GRN analyses, it is better to remove these data from the text to maintain clarity and focus on the relevant analyses.

RE6: We apologize as we did not explain our results enough. In fact, they are also an important aspect in order to develop the final model (Fig 4d-f) In short, the data from Δ loop and Δ lg3 and were used throughout the entire analysis e.g. when tracing the overbudding phenotype and other phenotypes using FDGENEA (Fig. 4/S10). We have added additional paragraphs describing these results.

Minor comments:

7) In Fig. 1c, the authors need to specify the number of observed filaments.

RE7: The number of observed filaments (n=100) was added in the Fig. 1 legend

8) To facilitate commenting and referencing, the authors should add page and line numbers in the text.

RE8: Page and line numbering was added.

9) In Fig. S5, there are two S5a. One of the two should be corrected to S5b.

RE9: Numbering was corrected in the supplement and references updated where appropriate.

10) The citation #47 mentioned on page 7 in relation to the NERD pathway components appears to be incorrect. It should be verified and corrected accordingly.

RE10: Reference numbering was corrected.

11) In Fig. S6a legend, the definitions of S1 through S4 (no, very few - few, few -medium, medium – many, and many – very many) are vague and require clarification to provide a better understanding of these terms.

RE11: These are just labels we use to describe clusters by their average number of predicted cleavage sites. We've added an explanatory sentence: "These labels were obtained by ranking the resulting k-means clusters by their centroid/mean number of predicted cleavage sites."

12) The text contains remnants of corrections, so it is necessary to make corrections.

RE12: We have gone over the text and revised it where appropriate.

REVIEWERS' COMMENTS:

Reviewer #1 (Remarks to the Author):

I would like to congratulate the authors for significantly improving the manuscript. The authors have addressed all my concerns and I am very pleased that the authors have put in tremendous effort to improve the readability of the text. I was able to finish reading the whole manuscript in one sitting and understand the bioinformatics. The newly added text also helps to put the whole analysis in perspective.

Here are just some minor points which require corrections:

1. Page 10 line 12: Please provide full name of SBP
2. Page 42 line 16: full form of LRT is missing. Although it is provided in the Materials & Methods, I imagine the legend text should appear first in the print.
3. Figure 3F: Is it possible to change the yellow text or increase the yellow shade? It is almost illegible on a white background. Also, only the black-coloured text was explained in the legend, what do the other 2 colours mean?
4. Legend of figure 3d: The opening sentence "Consistently, DEK1....2D-3D transition:" does not seem appropriate, but I leave it to the authors to decide.

The following references should appear as superscript in the text: 23, 79, 80

Reviewer #2 (Remarks to the Author):

The authors have effectively addressed all of my previous concerns. I am satisfied with the modifications made in the current manuscript.

Response to reviewers

Reviewer #1 (Remarks to the Author):

I would like to congratulate the authors for significantly improving the manuscript. The authors have addressed all my concerns and I am very pleased that the authors have put in tremendous effort to improve the readability of the text. I was able to finish reading the whole manuscript in one sitting and understand the bioinformatics. The newly added text also helps to put the whole analysis in perspective.

RE: Dear reviewer, we are deeply grateful that you once again took the time to thoughtfully and thoroughly work through our manuscript. We'd like to return the compliment and thank you for your effort as well as your very helpful comments and suggestions.

Here are just some minor points which require corrections:

1. Page 10 line 12: Please provide full name of SBP

RE1: done. Inserted "SQUAMOSA promoter binding protein-like (SBP)"

2. Page 42 line 16: full form of LRT is missing. Although it is provided in the Materials & Methods, I imagine the legend text should appear first in the print.

RE2: done. inserted "Likelihood-ratio test (LRT)" at first occurrence

3. Figure 3F: Is it possible to change the yellow text or increase the yellow shade? It is almost illegible on a white background. Also, only the black-coloured text was explained in the legend, what do the other 2 colours mean?

RE3: Sorry for the confusion. The color-code in 3F corresponds to the color-code of the 11 subnetworks which is used throughout the manuscript, figures and supplemental materials. I.e. yellow corresponds to enrichment of a term among target genes of subnetwork II.

We added grey shadows for the yellow text in Figure 3F to improve readability and adjusted the legend text as follows: "Text color code depicts subnetwork identity (i.e. subnetworks II, V and X) of (indirect) target gene. Black text corresponds to overall enrichment among target genes."

4. Legend of figure 3d: The opening sentence "Consistently, DEK1....2D-3D transition:" does not seem appropriate, but I leave it to the authors to decide.

RE4: Correct. We have changed the text as follows: "DEK1 calpain-dependent misregulation in the three subnetworks implementing the 2D-to-3D transition: misregulated genes in..."

The following references should appear as superscript in the text: 23, 79, 80

RE: done. Thanks for catching that.

Reviewer #2 (Remarks to the Author):

The authors have effectively addressed all of my previous concerns. I am satisfied with the modifications made in the current manuscript.

RE: Dear reviewer, thank you very much for reading the manuscript again and for your help in improving the manuscript. Your effort, helpful comments and suggestions are very much appreciated.